**Modeling of the chemistry in oxidation flow reactors with high initial NO**
Zhe Peng and Jose L. Jimenez
Cooperative Institute for Research in Environmental Sciences and Department of Chemistry, University of
Colorado, Boulder, CO 80309, USA
Correspondence to: J.L. Jimenez (jose.jimenez@colorado.edu)
**Abstract.** Oxidation flow reactors (OFRs) are increasingly employed in atmospheric chemistry research
because of their high efficiency of OH radical production from low-pressure Hg lamp emissions at both
185 and 254 nm (OFR185) or 254 nm only (OFR254). OFRs have been thought to be limited to studying
low-NO chemistry (where peroxy radicals ($RO_2$) react preferentially with $HO_2$) because NO is very rapidly
oxidized by the high concentrations of $O_3$, $HO_2$, and OH in OFRs. However, many groups are performing
experiments aging combustion exhaust with high NO levels, or adding NO in the hopes of simulating
high-NO chemistry (where $RO_2$ + NO dominates). This work systematically explores the chemistry in
OFRs with high initial NO. Using box modeling, we investigate the interconversion of N-containing
species and the uncertainties due to kinetic parameters. Simple initial injection of NO in OFR185 can
result in more $RO_2$ reacted with NO than with $HO_2$ and minor non-tropospheric photolysis, but only
under a very narrow set of conditions (high water mixing ratio, low UV intensity, low external OH
reactivity ($OHR_{ext}$), and initial NO concentration ($NO^{in}$) of tens to hundreds of ppb) that account for a
very small fraction of the input parameter space. These conditions are generally far away from
experimental conditions of published OFR studies with high initial NO. In particular, studies of aerosol
formation from vehicle emissions in OFR often used $OHR_{ext}$ and $NO^{in}$ several orders of magnitude higher.
Due to extremely high $OHR_{ext}$ and $NO^{in}$, some studies may have resulted in substantial non-tropospheric
photolysis, strong delay to $RO_2$ chemistry due to peroxynitrate formation, VOC reactions with $NO_3$
dominating over those with OH, and faster reactions of OH-aromatic adducts with $NO_2$ than those with
$O_2$, all of which are irrelevant to ambient VOC photooxidation chemistry. Some of the negative effects
are worst for alkene and aromatic precursors. To avoid undesired chemistry, vehicle emissions generally
need to be diluted by a factor of >100 before being injected into OFR. However, sufficiently diluted
vehicle emissions generally do not lead to high-NO chemistry in OFR, but are rather dominated by the
low-NO $RO_2$+$HO_2$ pathway. To ensure high-NO conditions without substantial atmospherically irrelevant
chemistry in a more controlled fashion, new techniques are needed.

## 1    Introduction

The oxidation of gases that are emitted into the atmosphere, in particular volatile organic compounds (VOCs), is one of the most important atmospheric chemistry processes (Haagen-Smit, 1952; Chameides et al., 1988). VOC oxidation is closely related to radical production and consumption (Levy II, 1971), $O_3$ production, and formation of secondary aerosols (Odum et al., 1996; Hoffmann et al., 1997; Volkamer et al., 2006; Hallquist et al., 2009), which have impacts on air quality and climate (Lippmann, 1991; Nel, 2005; Stocker et al., 2014).

Chemical reactors are critical tools for research of VOC oxidation. Oxidation reactions of interest often have typical timescales of hours to weeks. Studying these processes in ambient air can be confounded by dispersion and changes in ambient conditions, which often occur on similar timescales. Chemical reactors allow for decoupling of these two types of processes. Also, they should be able to simulate the different regimes of reactions occurring in the atmosphere, e.g., VOC oxidation under low and high-NO conditions (peroxy radical fate dominated by reaction with $HO_2$ or with NO) representing remote and urban areas, respectively (Orlando and Tyndall, 2012).

Large environmental chambers are a commonly used reactor type (Carter et al., 2005; Wang et al., 2011). They typically employ actinic wavelength (>300 nm) light sources (e.g., outdoor solar radiation and UV blacklights) to produce oxidants and radicals and have large volumes (on the order of several cubic meters or larger). However, the capability of generating sustained elevated levels of OH, the most important tropospheric oxidant, is usually limited in chambers, resulting in OH concentrations similar to those in the atmosphere ($10^6$–$10^7$ molecules cm$^{-3}$; Mao et al., 2009; Ng et al., 2010), and consequently, long simulation times (typically hours) to reach OH equivalent ages of atmospheric relevance (George et al., 2007; Kang et al., 2007; Carlton et al., 2009; Seakins, 2010; Wang et al., 2011). The partitioning of gases and aerosols to chamber walls (usually made of Teflon) in timescales of tens of minutes to hours makes it difficult to conduct very long experiments that simulate high atmospherically-relevant photochemical ages (Cocker et al., 2001; Matsunaga and Ziemann, 2010; Zhang et al., 2014; Krechmer et al., 2016). In addition, the long simulation times and large size of chambers and auxiliary equipment are logistically difficult for field deployment, and their cost limits the number of laboratories equipped with them.

Given the limitations of environmental chambers, a growing number of experimenters have instead employed oxidation flow reactors (OFRs). OFRs have a much smaller size (of the order of 10 L), efficiently generate OH via photolysis of $H_2O$ and/or $O_3$ by more energetic 185 and 254 nm photons from low-pressure Hg lamps, and overcome the abovementioned shortcomings of chambers due to a much shorter residence time (George et al., 2007; Kang et al., 2007, 2011; Lambe et al., 2011). Moreover, OFRs are able to rapidly explore a wide range of OH equivalent ages within a short period (~2 hr), during which significant changes of ambient conditions can usually be avoided in the case of field deployment (Ortega et al., 2016; Palm et al., 2016, 2017). Because of these advantages, OFRs have recently been widely used to study atmospheric chemistry, in particular secondary organic aerosol (SOA) formation and aging, in both the laboratory and the field (Kang et al., 2011; Li et al., 2013; Ortega et al., 2013,

2016; Tkacik et al., 2014; Palm et al., 2016).
In addition to experimental studies using OFRs, there has also been some progress in the
characterization of OFR chemistry by modeling. Li et al. (2015) and Peng et al. (2015) developed a box
model for OFR $HO_x$ chemistry that predicts measurable quantities [e.g., OH exposure ($OH_{exp}$, in
molecules $cm^{-3}$ s] and $O_3$ concentration (abbr. $O_3$ hereinafter, in ppm)] in good agreement with
experiments. This model has been used to characterize $HO_x$ chemistry as a function of $H_2O$ mixing ratio
(abbr. $H_2O$ hereinafter, unitless), UV light intensity (abbr. UV hereinafter, in photons $cm^{-2}$ $s^{-1}$), and
external OH reactivity [in $s^{-1}$, $OHR_{ext}=\sum k_i c_i$, i.e., the sum of the products of concentrations of externally
introduced OH-consuming species ($c_i$) and rate constants of their reactions with OH ($k_i$)]. Based on this
characterization, Peng et al. (2015) found that OH suppression, i.e., reduction of OH concentration
caused by $OHR_{ext}$, is a common feature under many typical OFR operation conditions. Peng et al. (2016)
systematically examined the relative importance of non-OH/non-tropospheric reactants on the fate of
VOCs over a wide range of conditions, and provided guidelines for OFR operation to avoid non-
tropospheric VOC photolysis, i.e., VOC photolysis at 185 and 254 nm.
In previous OFR modeling studies, $NO_x$ chemistry was not investigated in detail, since in such in
typical OFR experiments with large amounts of oxidants (e.g., OH, $HO_2$, and $O_3$), NO would be very
rapidly oxidized and thus unable to compete with $HO_2$ for reaction with peroxy radicals ($RO_2$). Li et al.
(2015) estimated an NO ($NO_2$) lifetime of ~0.5 (~1.5) s under a typical OFR condition. From these
estimates, OFRs processing ambient air or laboratory air without large addition of $NO_x$ were assumed
to be not suitable for studying oxidation mechanisms relevant to polluted conditions under higher NO
concentrations. OFRs have recently been used to conduct laboratory experiments with very high initial
$NO_x$ levels (Liu et al., 2015) and deployed to an urban tunnel, where $NO_x$ was high enough to be a major
OH reactant (Tkacik et al., 2014). The former study reported evidence for the incorporation of nitrogen
into SOA. Besides, OFRs have been increasingly employed to process emissions of vehicles, biomass
burning, and other combustion sources (Table 1), where NO can often be hundreds of ppm (Ortega et
al., 2013; Martinsson et al., 2015; Karjalainen et al., 2016; Link et al., 2016; Schill et al., 2016; Alanen et
al., 2017; Simonen et al., 2017). It can be expected that such a high NO input together with very high
VOC concentrations would cause a substantial deviation from good OFR operation conditions identified
in Peng et al. (2016). Very recently, $N_2O$ injection has been proposed by Lambe et al. (2017) as a way to
study oxidation of VOCs under high NO conditions in OFR. As more OFR studies at high $NO_x$ level are
conducted, there is growing need to understand the chemistry of N-containing species in OFRs and
whether it proceeds along atmospherically-relevant channels.
In this study, we present the first comprehensive model of OFR $NO_y$ chemistry. We extend the
model of Li et al. (2015) and Peng et al. (2015) by including a scheme for $NO_y$ species. Then this model
is used to investigate i) if an OFR with initial NO injection results in NO significantly reacting with $RO_2$
under any conditions, ii) if previously published OFR experiments with high initial NO concentrations
led to $RO_2$+NO being dominant in VOC oxidation without negative side effects (e.g., non-tropospheric
reactions), iii) how to avoid undesired chemistry in future studies. The results can provide insights into

 the design and interpretation of future OH-oxidation OFR experiments with large amounts of $NO_x$

injection.

## 2    Methods

The physical design of the OFR modeled in the present work, the chemical kinetics box model, and
the method of propagating and analyzing the parametric uncertainties on the model have already been
introduced previously (Kang et al., 2007; Li et al., 2015; Peng et al., 2015). We only provide brief
descriptions for them below.

### 2.1    Potential Aerosol Mass flow reactor

The OFR modeled in this study is the "Potential Aerosol Mass" (PAM) flow reactor, firstly
introduced by Kang et al. (2007). The PAM OFR is a cylindrical vessel with a volume of ~13 L, equipped
with low-pressure Hg lamps (model no. 82-9304-03, BHK Inc.) to generate 185 and 254 nm UV light.
This popular design is being used by many atmospheric chemistry research groups, particularly those
studying SOA (Lambe and Jimenez, 2017 and references therein). When the lamps are mounted inside
Teflon sleeves, photons at both wavelengths are transmitted and contribute to OH production ("OFR185
mode"). In OFR185, $H_2O$ photolyzed at 185 nm produces OH and $HO_2$, while $O_2$ photolyzed at the same
wavelengths results in $O_3$ formation. $O(^1D)$ is produced via $O_3$ photolysis at 254 nm and generates
additional OH through its reaction with $H_2O$. 185 nm lamp emissions can be filtered by mounting the
lamps inside quartz sleeves, leaving only 254 nm photons to produce OH ("OFR254 mode"). In this mode,
injection of externally formed $O_3$ is necessary to ensure OH production. As the amount of $O_3$ injected is
a key parameter under some conditions (Peng et al., 2015), we adopt the notation OFR254-X to denote
OFR254 experiments with X ppm initial $O_3$ ($O_{3,in}$). In this study, we investigate OFR experiments with NO
injected and thus utilize "OFR185-iNO" to describe the OFR185 mode of operation with initially (at the
reactor entrance) injected NO. The same terminology is used for the OFR254 mode. For instance, the
initial NO injection into OFR254-7 is denoted as OFR254-7-iNO.

### 2.2    Model description

The basic framework of the box model used in this study, a standard chemical kinetics model, is
the same as in Peng et al. (2015). Plug flow is assumed in the model, since approximately taking
residence time distribution into account leads to similar results under most conditions but at much
higher computational expense (Peng et al., 2015). In addition to the reactions in the model of Peng et
al. (2015), including all $HO_x$ reactions available in the JPL Chemical Kinetic Data Evaluation (Sander et al.,
2011), all gas-phase $NO_y$ reactions available in the JPL database except those of organic nitrates and
peroxynitrates are also considered in the current reaction scheme. An updated JPL evaluation was
published recently (Burkholder et al., 2015), with slightly different (~20%) rate constants for
$NO_2+HO_2+M{\rightarrow}HO_2NO_2+M$ and $NO_2+NO_3{\rightarrow}N_2O_5$. The updated rate constants only result in changes of
~10–20% of the concentrations of the species directly consumed/produced by these reactions. These
changes are smaller than the parametric uncertainties of the model (see Section 3.1.3). For other
species, concentration changes are negligible. $HO_2NO_2+M{\rightarrow}HO_2+NO_2+M$ and $N_2O_5+M{\rightarrow}NO_2+NO_3+M$,
are also included in the scheme, with kinetic parameters from the IUPAC Task Group on Atmospheric
Chemical Kinetic Data Evaluation (Ammann et al., 2016). As in Peng et al. (2015, 2016), $SO_2$ is used as a
surrogate of external OH reactants (e.g., VOCs). $NO_y$ species, although also external OH reactants, are
explicitly treated in the model and *not* counted in $OHR_{ext}$ in this work. Therefore, $OHR_{ext}$ stands for *non-*
$NO_y$ $OHR_{ext}$ only hereinafter, unless otherwise stated.
Also, particle-phase chemistry and physical and chemical interactions of gas-phase species with
particles are not considered in this study. We have made this assumption because:
i)  The presence of aerosols has typically negligible impacts on the gas-phase chemistry of radicals,
$NO_x/NO_y$, and OH reactants studied here. Condensational sink (CS) of ambient aerosols can
rarely exceed 1 $s^{-1}$ even in polluted areas and is usually 1-3 orders of magnitude lower
(Donahue et al., 2016; Palm et al., 2016). Thus, even under the assumption of unity uptake
coefficient, CS cannot compete with $OHR_{ext}$ (usually on the order of 10 $s^{-1}$ or higher) in OH loss.
Uptake of NO onto aerosols only occurs through the reaction with $RO_2$ on particle surface
(Richards-Henderson et al., 2015), which is formed very slowly (see below) compared to gas-
phase $HO_x$ and $NO_x$ chemistry. Uptake of $HO_2$, $O_3$, $NO_3$ etc. is even more unlikely to be of
importance due to lower uptake coefficients (Moise and Rudich, 2002; Moise et al., 2002;
Hearn and Smith, 2004; Lakey et al., 2015). Combustion exhausts can have high aerosol
loadings with condensational sinks on the order of $10^2$–$10^3$ $s^{-1}$ (Matti Maricq, 2007). Even if
these exhausts are directly injected into the reactor without any pre-treatment, uptake onto
the particles still cannot play a major role in the fate of gas-phase radical and $NO_x$ species, since
VOCs and $NO_x$ in raw exhausts, which are proportionally orders-of-magnitude higher, still
dominate the fate of oxidants. Dilution of combustion emissions simultaneously lowers
condensational sinks and the sinks of oxidants due to chemical reactions, with their relative
importance remaining the same as in undiluted emissions.
ii)  Gas-phase radical and $NO_x/NO_y$ species only has limited impacts on OA *chemistry* in this study.
Heterogeneous oxidation of OA by OH is generally slow. Significant OA loss due to
heterogeneous oxidation can only be seen at photochemical ages as high as weeks (Hu et al.,
2016). The enhancement of heterogeneous oxidation due to NO is remarkable only at OH
concentration close to the ambient values but not at typical values in OFR (Richards-Henderson
et al., 2015).
It is well known that the aerosol concentration can have a major impact on the physical uptake of
semivolatile and low-volatility gas-phase species. However this process is not explicitly modeled in this
study.
As $OHR_{ext}$ plays a major and even dominant role in OH loss, it is an important approximation that
the *real* $OHR_{ext}$ decay (due to not only primary VOC oxidation and subsequent oxidation of higher
generation products, but also wall loss, partitioning to the particle phase, reactive uptake etc.) is
surrogated by that of $SO_2$ (see Fig. S2 of Peng et al. 2015). Gas-phase measurements in literature
laboratory studies revealed that there is a large variability of the evolution of total $OHR_{ext}$ during
oxidation of primary VOCs and subsequent oxidation of their intermediate products, depending on the
type of precursors (Nehr et al., 2014; Schwantes et al., 2017). This variability is obviously mainly due to
the formation of different types and amounts of oxidation intermediates/products contributing to
$OHR_{ext}$. This variation is highly complex due to the large number of possible oxidation intermediates and
the limited knowledge of detailed higher-generation mechanisms, and thus is difficult to accurately
capture even if modeling with a mechanism as explicit as Master Chemical Mechanism is performed
(Schwantes et al., 2017). Therefore, it is justified to use a lumped surrogate to model the $OHR_{ext}$ decay
for simplicity and efficiency. This approximation is a major contributor to uncertainty of our model. The
uncertainties due to both the types of oxidation intermediates/products.

A residence time of 180 s and typical temperature (295 K) and atmospheric pressure (835 mbar)

in Boulder, CO, USA are assumed for all model cases. The lower-than-sea level pressure only leads to
minor differences in the outputs (Li et al., 2015). We explore physical input cases evenly spaced in a
logarithmic scale over very wide ranges: $H_2O$ of 0.07%–2.3%, i.e., relative humidity (RH) of 2–71% at
295 K; 185 nm UV of $1.0 \times 10^{11}$–$1.0 \times 10^{14}$ and 254 nm UV of $4.2 \times 10^{13}$–$8.5 \times 10^{15}$ photons $cm^{-2}$ $s^{-1}$; $OHR_{ext}$ of
1–16000 $s^{-1}$; $O_{3,in}$ of 2.2–70 ppm for OFR254; initial NO mixing ratio ($NO^{in}$) from 10 ppt to 40 ppm.
Besides, conditions with $OHR_{ext}=0$ are also explored. UV at 254 nm is estimated from that at 185 nm
according to the relationship determined by Li et al. (2015). Several typical cases within this range as
well as their corresponding 4 or 2-character labels (e.g., MM0V and HL) are defined in Table 2. Literature
studies are modeled by adopting all reported parameters (e.g., residence time, $H_2O$, and $O_{3,in}$) and
estimating any others that may be needed (e.g., UV) from the information provided in the papers.

In this study, OH equivalent ages are calculated under the assumption of an ambient OH

concentration of $1.5 \times 10^6$ molecules $cm^{-3}$ (Mao et al., 2009). Conditions leading to a ratio of $RO_2$ reacted
with NO over the entire residence time [$r(RO_2+NO)$] to that with $HO_2$ [$r(RO_2+HO_2)$] larger than 1 are
regarded as "high NO" (under the assumption of constant $OHR_{ext}$ from VOCs, see Section S1 for more
details), where [$r(X)$] is the total reactive flux for reaction X over the entire residence time. $F185_{exp}/OH_{exp}$
and $F254_{exp}/OH_{exp}$ are used as measures of the relative importance of VOC photolysis at 185 and 254
nm to their reactions with OH, respectively [$F185_{exp}$ ($F254_{exp}$) are 185 (254) nm photon flux exposure,
i.e., product of 185 (254) nm photon flux and time]. Readers may refer to Figs. 1 and 2 of Peng et al.
(2016) for the determination of the relative importance of non-tropospheric (185 and 254 nm)
photolysis of individual VOCs. Although the relative importance of non-tropospheric photolysis depends
on individual VOCs, in the present work, we set criteria on $F185_{exp}/OH_{exp}<3 \times 10^3$ cm/s and
$F254_{exp}/OH_{exp}<4 \times 10^5$ cm/s to define "good" conditions and $F185_{exp}/OH_{exp}<1 \times 10^5$ cm/s and
$F254_{exp}/OH_{exp}<1 \times 10^7$ cm/s (excluding good conditions) to define "risky" conditions. Conditions with
higher $F185_{exp}/OH_{exp}$ or $F254_{exp}/OH_{exp}$ are defined as "bad". Under good conditions, photolysis of most
VOCs has a relative contribution <20% to their fate; under bad conditions, non-tropospheric photolysis
is likely to be significant in all OFR experiments, since it can hardly be avoided for oxidation
intermediates, even if the precursor(s) does not photolyze at all. Under risky conditions, some species
photolyzing slowly and/or reacting with OH rapidly (e.g., alkanes, aldehydes, and most biogenics) still
have a relative contribution of photolysis <20% to their fates, while species photolyzing more rapidly
and/or reacting with OH more slowly (e.g., aromatics and other highly conjugated species and some
saturated carbonyls) will undergo substantial non-tropospheric photolysis. Note that these definitions
are slightly different than in Peng et al. (2016). All definitions of the types of conditions are summarized
in Table 3.
**2.3    Uncertainty analysis**

We apply the same method as in Peng et al. (2014, 2015) to calculate and analyze the output

uncertainties due to uncertain kinetic parameters in the model. Random samples following log-normal
distributions are generated for all rate constants and photoabsorption cross sections in the model using
uncertainty data available in the JPL database (Sander et al., 2011) or estimated based on IUPAC data
(Ammann et al., 2016). Then, Monte Carlo Uncertainty Propagation (BIPM et al., 2008) is performed for
these samples through the model to obtain the distributions of outputs. Finally, we compute squared
correlation coefficients between corresponding input and output samples and apportion the relative
contributions of individual kinetic parameters to the output uncertainties based on these coefficients
(Saltelli et al., 2005).
**3    Results and discussion**

In this section, we study the $NO_y$ chemistry in OFR while considering relevant experimental issues.

Based on these results, we propose some guidelines for OFR operation for high-NO OH oxidation of
VOCs.
**3.1    $NO_y$ chemistry in typical OFR cases with initial NO injection**

NO was thought to be unimportant (i.e., unable to significantly react with $RO_2$) in OFRs with initial

NO injection (OFR-iNO) based on the argument that its lifetime is too short due to large amounts of $O_3$
OH, and $HO_2$ to compete with $RO_2+HO_2$ (Li et al., 2015). We evaluate this issue below by calculating NO
effective lifetime ($\tau_{NO}$, in s), defined as NO exposure ($NO_{exp}$, in molecules $cm^{-3}$ s) divided by initial NO
concentration, under various conditions. This definition cannot effectively capture the true NO average
lifetime if it is close to or longer than the residence time. In this case, $\tau_{NO}$ close to the residence time
will be obtained, which is still long enough for our characterization purposes.
**3.1.1    OFR185-iNO**

In OFR185-iNO, NO is *not* oxidized extremely quickly under *all* conditions. For instance, under a

typical condition in the midrange of the phase space shown in Fig. 1a, $\tau_{NO}$ ~13 s. This lifetime is much
shorter than the residence time, but long enough for $OH_{exp}$ to reach ~$3x10^{10}$ molecules $cm^{-3}$ s, which is
equivalent to an OH equivalent age of ~6 hrs. Such an OH equivalent age is already sufficient to allow
some VOC processing and even SOA formation to occur (Lambe et al., 2011; Ortega et al., 2016). Within
$\tau_{NO}$, NO suppresses $HO_2$ through the reaction $NO+HO_2 \rightarrow NO_2+OH$, leading to $NO_{exp}/HO_{2exp}$ of ~700 during
this period, high enough for $RO_2$ to dominantly react with NO. Meanwhile, $NO+HO_2 \rightarrow NO_2+OH$ enhances
OH production, which helps $OH_{exp}$ build up in a relatively short period. In addition, non-tropospheric
photolysis of VOCs at 185 and 254 nm is minor ($F185_{exp}/OH_{exp}$ ~ 600 cm/s, Fig. 1a), because of enhanced
OH production and moderate UV. Therefore, such an OFR condition may be of some interest for high-
NO VOC oxidation. We thus analyze the $NO_y$ chemistry in OFR185-iNO in more detail below, by taking
the case shown in Fig. 1a as a representative example.

In OFR185-iNO, $HO_x$ concentrations are orders-of-magnitude higher than in the atmosphere

while the amount of $O_3$ produced is relatively small during the first several seconds after the flow enters
the reactor. As a result, NO is not oxidized almost exclusively by $O_3$ as in the troposphere, but also by
OH and $HO_2$ to form HONO and $NO_2$, respectively (Fig. 1a). The large concentration of OH present then
oxidizes HONO to $NO_2$, and $NO_2$ to $HNO_3$. Photolysis only plays a negligible role in the fate of HONO and
$NO_2$ in OFRs, in contrast to the troposphere, where it is the main fate of these species. This is because
the reactions of HONO and $NO_2$ with OH are greatly accelerated in OFR compared to those in the
troposphere, while photolysis not (Peng et al., 2016). The interconversion between $NO_2$ and $HO_2NO_2$ is
also greatly accelerated (Fig. 1a), since a large amount of $HO_2$ promotes the formation of $HO_2NO_2$,
whose thermal decomposition and reaction with OH in turn enhance the recycling of $NO_2$. Though not
explicitly modeled in this study, $RO_2$ are expected to undergo similar reactions with $NO_2$ to form
reservoir species, i.e., peroxynitrates (Orlando and Tyndall, 2012). Peroxynitrates that decompose on
timescales considerably longer than OFR residence times may serve as effectively permanent $NO_y$ sinks
in OFRs (see Section 3.4.1).

Interestingly but not surprisingly, the $NO_y$ chemistry shown in Fig. 1a is far from temporally

uniform during the OFR residence time (Fig. S1a). Within $\tau_{NO}$, NO undergoes an e-fold decay as it is
rapidly converted into $NO_2$ and HONO, whose concentrations reach maxima around that time. After
most NO is consumed, HONO and $NO_2$ also start to decrease, but significantly more slowly than NO,
since they do not have as many and efficient loss pathways as NO. The reaction of OH with HONO, the
dominant fate of HONO, is slower than that with NO (Fig. 1a). The net rate of the $NO_2$-to-$HO_2NO_2$
conversion becomes low because of the relatively fast reverse reaction (Fig. 1a). Besides, the total loss
of $NO_2$ is partially offset by the production from HONO. The generally stable concentrations of HONO
and $NO_2$ (Fig. S1a) result in their respective reaction rates with OH that are comparable during and after
$\tau_{NO}$ (Fig. 1a), as OH variation is also relatively small during the entire residence time (Fig. S1b). However,
the $NO_2$-to-$HO_2NO_2$ conversion after $\tau_{NO}$ is much faster than during it (Fig. 1a), resulting from
substantially decreased NO and $HO_2$ concomitantly increasing >1 order of magnitude after $\tau_{NO}$ (Fig.
S1a,b). $HNO_3$ and $HO_2NO_2$, which are substantially produced only after $NO_2$ is built up, have much higher
concentrations later than within $\tau_{NO}$.

Under other OFR185-iNO conditions than in Fig. 1a, the major reactions interconverting $NO_y$

species are generally the same, although their relative importance may vary. At lower $NO^{in}$, the
perturbation of $HO_x$ chemistry caused by $NO_y$ species is smaller. Effects of $NO^{in}$ less than 1 ppb (e.g.,
typical non-urban ambient concentrations) are generally negligible regarding $HO_x$ chemistry. Regarding
$NO_y$ species, the pathways in Fig. 1a are still important under those conditions. At higher $NO^{in}$ (e.g., >1
ppm), one might expect $NO_3$ and $N_2O_5$ to play a role (as in OFR254-iNO; see Section 3.1.2 below), since
high $NO_y$ concentrations might enhance self/cross reactions of $NO_y$. However, this would not occur
unless OH production is high, since relatively low $O_3$ concentrations in OFR185-iNO cannot oxidize $NO_2$
to $NO_3$ rapidly. Also, a large amount of $NO_y$ can lead to significant OH suppression. That would in turn

slow down the $NO_3$ production from $HNO_3$ by OH. This is especially true when an OFR is used to oxidize the output of highly concentrated sources (e.g., from vehicle exhausts). When sources corresponding to $OHR_{ext}$ of thousands of $s^{-1}$ and $NO^{in}$ of tens of ppm are injected into OFR185 (Fig. 1b), they essentially inhibit active chemistry except NO consumption, as all subsequent products are much less abundant compared to remaining NO (Fig. S1c).

### 3.1.2 OFR254-iNO

The ppm-level $O_{3,in}$ used in the OFR254-iNO mode of operation has a strong impact on its $NO_y$ chemistry. An $O_{3,in}$ of 2.2 ppm (lowest in this study) is already enough to shorten $\tau_{NO}$ to ~1 s, preventing NO from playing a role in the chemistry under most explored conditions. The reaction fluxes under a typical $O_{3,in}$ of 7 ppm are shown in Fig. 1c. A reactive flux from $NO+O_3 \rightarrow NO_2$ makes the reaction of NO with other oxidants (OH, $HO_2$ etc.) negligible. The $HNO_3$ production pathway from $NO_2$ is similar to that in OFR185-iNO. The interconversion between $NO_2$ and $HO_2NO_2$ is also fast over the residence time, and even faster than in OFR185-iNO during $\tau_{NO}$, since a high concentration of $O_3$ also controls the OH-$HO_2$ interconversion and makes $HO_2$ more resilient against suppression due to high NO (Fig. S1f; Peng et al., 2015). A major difference in the $NO_y$ chemistry in OFR254-iNO (Fig. 1c) compared to OFR185-iNO (Fig. 1a) is significant $NO_3/N_2O_5$ chemistry due to high $O_3$ in OFR254-iNO, which accelerates the oxidation of $NO_2$ to $NO_3$. Interconversion between $NO_2+NO_3$ and $N_2O_5$ also occurs to a significant extent because of high $NO_2$. Under the conditions of Fig. 1c, $NO_3$ can also be significantly consumed by $HO_2$. Unlike OFR185-iNO, OFR254-iNO can substantially form $NO_3$ from $HNO_3$ under conditions that are not on the extremes of the explored physical condition space, e.g., at higher UV and lower $NO^{in}$ (e.g., Fig. S2). In the case of very high $NO^{in}$ (equal to or higher than $O_{3,in}$), all $O_3$ can be rapidly destroyed by NO. As a consequence, OH production is shut down and these cases are of little practical interest (Fig. S3h).

### 3.1.3 Uncertainty analysis

The results of uncertainty propagation confirm that the output uncertainties due to uncertain kinetic parameters are relatively low compared to other factors (e.g., non-plug flow in OFR; Peng et al., 2015) and the overall model accuracy compared to experimental data (a factor of 2–3; Li et al., 2015). For OFR185-iNO, NO, $NO_3$, and OH exposures have relative uncertainties of ~0–20%, ~40–70%, and ~15–40%, respectively. The uncertainties in OH exposure are very similar to those in the cases without $NO_x$ (Peng et al., 2015). The contribution of $NO_y$ reactions to $OH_{exp}$ uncertainty is negligible, except for some contribution of $OH+NO \rightarrow HONO$ in a few cases with high $NO^{in}$ (Fig. 2). The uncertainties on $NO_{exp}$ are dominated by the reactions producing $HO_x$ and $O_3$, i.e., the major consumers of NO. For $NO_3$ exposure, a few major production and loss pathways (e.g., $NO_2+NO_3 \rightarrow N_2O_5$, $N_2O_5 \rightarrow NO_2+NO_3$, and $HO_2+NO_3 \rightarrow OH+NO_2+O_2$) dominate its uncertainties. OFR254-iNO has a simpler picture of parametric uncertainties in terms of composition. $O_3$ controls the NO oxidation under most conditions and this reaction contributes most of output uncertainties for NO exposures. $HO_2+NO_3 \rightarrow OH+NO_2+O_2$ dominates the uncertainty on $NO_3$ exposure. The levels of those uncertainties are lower than in OFR185-iNO (<2% for NO exposure; <60% in all cases and <25% in most cases for $NO_3$ exposure). Thus, model uncertainties in OFR254-iNO are not shown in detail.

## 3.2    Different conditions types

Having illustrated the main $NO_y$ chemical pathways for typical cases, we present the results of the exploration of the entire physical parameter space (see Section 2.2). Note that the explored space is indeed very large and gridded logarithmically uniformly in every dimension. Therefore, the statistics of the exploration results can be useful to determine the relative importance of the conditions types defined in Section 2.2 and Table 3.

It has been shown that during $\tau_{NO}$, $RO_2$ can react dominantly with NO (Section 3.1.1), while to determine if a condition is high-NO (see Table 3), the entire residence time is considered. This is done because for VOC oxidation systems of interest, there will be significant oxidation of the initial VOC and its products under low-NO conditions, if $\tau_{NO}$ is shorter than the reactor residence time. After most NO is consumed, the longer the remaining residence time, the more $RO_2$ will react with $HO_2$ and the more likely that an input condition is classified as low-NO. For a condition to be high-NO, a significantly long $\tau_{NO}$ is required. Figure 3 shows the fractional occurrence distribution of good/risky/bad conditions in the entire explored condition space over logarithm of $r(RO_2+NO)/r(RO_2+HO_2)$, which distinguishes high- and low-NO conditions. In OFR254-iNO, $\tau_{NO}$ is so short that no good high-NO condition is found in the explored range in this study (Fig. 3a). A fraction of explored conditions are bad high-NO. These conditions result from a full consumption of $O_3$ by NO. Then very little $HO_x$ is produced (right panels in Fig. S3h), but the fate of any $RO_2$ formed is dominated by $RO_2+NO$ (right panels in Fig. S3i). However, also due to negligibly low OH concentration, little $RO_2$ is produced and non-tropospheric photolysis of VOCs is also substantial compared to their reaction with OH under these conditions, classifying all of them as "bad" (Fig. 3a).

In OFR185-iNO, in addition to the typical case shown in Fig. 1a, many other cases have a $\tau_{NO}$ of ~10 s or longer (Figs. S3b and S4), which allow the possibility of high-NO conditions. Indeed, ~1/3 of explored conditions in OFR185-iNO with a residence time of 3 min are high-NO (Fig. 3b). Most of these high-NO conditions are also classified as bad, similar with those in OFR254-iNO. More importantly, in contrast to OFR254-iNO, good and risky high-NO conditions also comprise an appreciable fraction of the OFR185-iNO conditions. It is easily expected that very high $OHR_{ext}$ and $NO^{in}$ lead to bad high-NO conditions (all panels in Fig. 4), since they strongly suppress $HO_x$, which yields bad conditions and in turn keep NO destruction relatively low. Besides, the occurrence of bad high-NO conditions is reduced at high UV (bottom panels in Fig. 4), which can be explained by lowered NO due to high $O_3$ production and fast OH reactant loss due to high OH production. Good high-NO conditions are rare in the explored space. They are only 1.1% of total explored conditions (Fig. 3b) and present under very specific conditions, i.e., higher $H_2O$, lower UV, lower $OHR_{ext}$, and $NO^{in}$ of tens to hundreds of ppb (Figs. 4 and S5). Since a very high NO can suppress OH, to obtain both a significant NO level and a good conditions, $NO^{in}$ can only be tens to hundreds of ppb. As $NO^{in}$ is lower and OH is higher than under bad high-NO conditions, UV should be lower than bad high-NO conditions to keep a sufficiently long presence of NO. Thus, UV at 185 nm for good high-NO conditions are generally lower than $10^{12}$ photons $cm^{-2}$ $s^{-1}$ (Fig. S5). In addition, a low $OHR_{ext}$ (generally <50 $s^{-1}$) and a higher $H_2O$ (the higher the better, although there is

no apparent threshold) are also required for good high-NO conditions (Fig. S5), as Peng et al. (2016) pointed out. Risky high-NO conditions often occur between good and bad high-NO conditions, e.g., at lower $NO^{in}$ than bad conditions (e.g., Cases ML, MM, HL, and HM in Fig. 4, see Table 2 for the typical case label code), at higher $OHR_{ext}$ and/or $NO_{in}$ than good conditions (e.g., Cases ML and MM), and at lower $H_2O$ than good conditions (e.g., Case LL).

The trend of the distributions of good, risky, and bad low-NO conditions is generally in line with the analysis in Peng et al. (2016). For low-NO conditions, $NO_y$ species can be simply regarded as external OH reactants, as in Peng et al. (2016). As $H_2O$ decreases and/or $OHR_{ext}$ or $NO^{in}$ increases, a low-NO condition becomes worse (good→risky→bad) (Figs. 4 and 5). In OFR185-iNO, increasing UV generally makes a low-NO condition better because of an OH production enhancement (Fig. 4); while in OFR254-iNO, increasing UV generally makes a low-NO condition worse (Fig. 5), since at a higher UV, more $O_3$ is destroyed and the resilience of OH to suppression is reduced.

As discussed above, the fraction of high-NO conditions also depends on OFR residence time. A shorter residence time is expected to generally lead to a larger fraction of high-NO conditions, since the time spent in the reaction for $t > \tau_{NO}$ is significantly smaller. Thus, we also investigate an OFR185-iNO case with a residence time of 30 s. In Fig. 3b, compared to the case with a residence time of 3 min, the distributions of all condition types (good/risky/bad) of the 30 s residence time case shift toward higher $r(RO_2+NO)/r(RO_2+HO_2)$. Nevertheless, shortening the residence time also removes the period when the condition is better (i.e., less non-tropospheric photolysis), when external OH reactants have been partially consumed and OH suppression due to $OHR_{ext}$ has been reduced later in the residence time. As a result, the fractions of good and risky conditions decrease (Fig. 3b). With the two effects (higher $r(RO_2+NO)/r(RO_2+HO_2)$ and more significant non-tropospheric photolysis) combined, the fraction of good high-NO conditions increases by a factor of ~3. An even shorter residence time does not result in a larger good high-NO fraction, since the effect of enhancing non-tropospheric photolysis is even more apparent.

**3.3    Effect of non-plug flow**

We performed model runs where the only change with respect to our box model introduced in Section 2.2 is that the plug-flow assumption is replaced by the residence time distribution (RTD) measured by Lambe et al. (2011) (also see Fig. S8 of Peng et al. (2015)). The chemistry of different air parcels with different residence times is simulated by our box model and outputs are averaged over the RTD. Lateral diffusion between different air parcels is neglected in these simulations.

$OH_{exp}$ calculated from the mode with RTD ($OH_{exp,RTD}$) is higher than that calculated from the plug-flow model ($OH_{exp,PF}$) in both OFR185-iNO and OFR254-iNO (Table 4 and Fig. S6). Under most explored conditions deviations are relatively small, which leads to an overall positive deviation of $OH_{exp,RTD}$ from $OH_{exp,PF}$ by ~x2 (within the uncertainties of the model and its application to real experimental systems). For OFR185-iNO, most conditions (~90%) in the explored space lead to <x3 differences between $OH_{exp,PF}$ and $OH_{exp,RTD}$, while for a small fraction of cases the differences can be larger (Fig. S6). The larger deviations are mainly present at high UV, $OHR_{ext}$, and $NO^{in}$, where conditions are generally "bad" and in

which experiments are of little atmospheric relevance. Under these specific conditions, external OH
reactants and $NO_y$ can be substantially destroyed for the air parcels with residence times longer than
the average, while this is not the case for the average residence time. This feature was already described
by Peng et al. (2015) (see Fig. S10 of that study). Although only non-$NO_y$ external OH reactants were
considered in that study, the results are the same. In the present study, a higher upper limit of the
explored $OHR_{ext}$ range (compared to Peng et al., 2015, due to trying to simulate extremely high $OHR_{ext}$
used in some recent literature studies) large amounts of $NO_y$ and cause somewhat larger deviations. In
OFR254-iNO, OH is less suppressed at high $OHR_{ext}$ and $NO^{in}$ than in OFR185-iNO because of high $O_3$
(Peng et al., 2015), $OH_{exp,RTD}$ deviations from $OH_{exp,PF}$ are also smaller (Table 4).

Based on the outputs of the model with RTD, similar mapping of the physical input space as Figs.

4 and 5 can be done (Figs. S7 and S8). Overall, the mapping of the RTD model results is very similar with
that of the plug-flow model. The conditions appear to be only slightly better in a few places of the
explored space than those from the plug-flow model, which can be easily explained by the discussions
above. Besides, the mapping in Figs. S7 and S8 also appear to be slightly more low-NO, for the same
reasons discussed above. After NO is destroyed at long residence times, $HO_2$, suppressed by NO, also
recovers as OH. $r(RO_2+NO)/r(RO_2+HO_2)$ is obviously expected to be smaller than in the plug-flow model
in general.

Note that most conditions that appear to be better in the RTD model results are already

identified as bad by the plug-flow model. Those conditions look slightly better only because of their
better *RTD-averaged* $F185_{exp}/OH_{exp}$ and $F254_{exp}/OH_{exp}$. However, each of those cases is actually
composed of both a better part at longer residence times and also a worse part at shorter residence
times. Under those conditions, the reactor simultaneously works in two distinct regimes, one of which
is bad due to heavy OH suppression. Such conditions are obviously not desirable for OFR operation.
**3.4    Possible issues related to high-$NO_x$ levels**

In the discussion above, we focused on obtaining high-NO conditions and considered only one

experimental issue (non-tropospheric photolysis) that had been previously investigated in Peng et al.
(2016) and is not specific for experiments with high NO injection. We discuss additional potential
reasons why the OFR-iNO chemistry can deviate strongly from tropospheric conditions, as specifically
related to high-$NO_x$ level in this subsection.
**3.4.1  $NO_2$**

$NO_2$ reacts with $RO_2$ to form peroxynitrates, generally regarded as reservoir species in the

atmosphere as most of them thermally decompose very quickly compared to atmospheric time scales.
However, in OFRs, with residence times on the order of minutes, some peroxynitrates may no longer be
considered as fast decomposing. This is especially true for acylperoxy nitrates, whose lifetimes can be
hours at room temperature (Orlando and Tyndall, 2012). Acylperoxy nitrates are essentially sinks instead
of reservoirs in OFRs for both $NO_2$ and $RO_2$. $RO_2$ is estimated to be as high as several ppb in OFRs by our
model (e.g., ~6 ppb $RO_2$ in OFR185 at $H_2O$=1%, UV at 185 nm=1x10$^{13}$ photons cm$^{-2}$ s$^{-1}$, $OHR_{ext}$=1000 s$^{-1}$,
and $NO^{in}$=0), while high-NO experiments can yield far higher $NO_2$. If all $RO_2$ were acylperoxy, the $RO_2$
chemistry could be rapidly shut down by $NO_2$, as rate constants of these $RO_2 + NO_2$ reactions are around
$10^{-11}$ $cm^3$ molecule$^{-1}$ s$^{-1}$ (Orlando and Tyndall, 2012). Nevertheless, acylperoxy nitrates are not expected
to typically be the dominant component of peroxynitrates, since acyl radicals are not a direct oxidation
product of most common VOCs and can only be formed after several steps of oxidation (Atkinson and
Arey, 2003; Ziemann and Atkinson, 2012). Most alkylperoxy nitrates retain their short-lived reservoir
characteristics in OFRs due to their relatively short thermal decomposition time scales (on the order of
0.1 s; Orlando and Tyndall, 2012). Even so, OFR experiments can be seriously hampered at extremely
high $NO_2$. If $NO_2$ reaches ppm levels, the equilibrium between $RO_2+NO_2$ and alkylperoxy nitrate
($RO_2+NO_2 \leftrightarrow RO_2NO_2$) is greatly shifted toward the alkylperoxy nitrate side, as the forward and reverse
rate constants are on the order of $10^{-12}$ $cm^3$ molecule$^{-1}$ s$^{-1}$ and 1 s$^{-1}$, respectively (Orlando and Tyndall,
2012). This results in a substantial decrease in effective $RO_2$ concentration, or in other words, a
substantial slow-down of $RO_2$ chemistry.

Parts per million levels of $NO_2$ may impose an additional experimental artifact in the oxidation

chemistry of aromatic precursors. OH-aromatic adducts, i.e., the immediate products of aromatic
oxidation by OH, undergo addition of $O_2$ and $NO_2$ at comparable rates under ppm levels of $NO_2$ (rate
constants of the additions of $O_2$ and $NO_2$ are on the order of $10^{-16}$ and $10^{-11}$ molecules $cm^{-3}$ s$^{-1}$,
respectively ;Atkinson and Arey, 2003). However, only the former addition is atmospherically relevant
(Calvert et al., 2002). Liu et al. (2015) performed OFR254-iNO experiments with toluene over a range of
$NO^{in}$ of 2.5–10 ppm, encompassing the NO concentration range at which the reactions of OH-toluene
adduct with $O_2$ and with $NO_2$ are of equal importance (~5 ppm; Atkinson and Arey, 2003). This suggests
that nitroaromatics, whose formation was reported in the study of Liu et al. (2015), might have been
formed in substantial amounts in that study through the addition of $NO_2$ to the OH-toluene adduct.
**3.4.2 $NO_3$**

As discussed in Section 3.1, $NO_3$ can be formed in significant amounts in OFRs with high NO

injection. Although $NO_3$ is also present in the atmosphere, especially during nighttime, significant VOC
oxidation by both OH and $NO_3$ results in more complex chemistry that may complicate the
interpretation of experimental results. $NO_3$ oxidation-only OFR has been previously realized
experimentally via thermal dissociation of injected $N_2O_5$ (Palm et al., 2017). We discuss below how to
avoid significant VOC oxidation by $NO_3$ and achieve OH-dominated VOC oxidation in OFRs with high NO
injection.

If $NO_{3exp}/OH_{exp} > 0.1$, $NO_3$ can be a competitive reactant for biogenic alkenes and dihydrofurans,

which have a C=C bond for $NO_3$ addition, and phenols, which have activated hydroxyl for fast hydrogen
abstraction by $NO_3$ (Atkinson and Arey, 2003), while for lower $NO_{3exp}/OH_{exp}$, OH is expected to dominate
the oxidation of all VOCs, as shown in Fig. 6. Oxidation for VOCs without alkene C=C bonds and phenol
hydroxyl (such as alkanes and (alkyl)benzenes) is dominated by OH unless $NO_{3exp}/OH_{exp} > 1000$. Despite
its double bond, ethene reacts as slowly with $NO_3$ as alkanes, likely due to lack of alkyl groups enriching
electron density on the C=C bond, which slows $NO_3$ addition. We calculate $NO_{3exp}/OH_{exp}$ for OFR185-
iNO and OFR254-iNO and plot histograms of this ratio in Fig. 6. Many experimental conditions lead to
high enough $NO_{3exp}/OH_{exp}$ that $NO_3$ is a competitive sink for alkenes, while only under very extreme
conditions can $NO_3$ be a competitive sink for species without C=C bonds. High-NO conditions in OFR185-
iNO have lower $NO_{3exp}/OH_{exp}$ ($\sim 10^{-2}$–$10^2$) than in OFR254-iNO ($\sim 10^1$–$10^5$) (Figs. 6 and S3d,g,j). This
difference in $NO_{3exp}/OH_{exp}$ is due to the different levels of $O_3$ in the two modes, as high $O_3$ promotes
$NO_2$-to-$NO_3$ oxidation. Note that low-NO conditions in both OFR185-iNO and OFR254-iNO can also reach
high $NO_{3exp}/OH_{exp}$ as some high-NO conditions have. This is because in OFR185-iNO a large part of $NO_3$
is formed by OH oxidation, resulting in $NO_{3exp}/OH_{exp}$ being largely influenced by $NO^{in}$ but not by other
factors mainly governing OH (Fig. S3d); and under low-NO conditions in OFR254-iNO, $NO_3$ can form
rapidly from $NO_2+O_3$, while OH can be heavily suppressed by high $OHR_{ext}$ (Fig. S3g,j).

Most of the species shown in Fig. 6 are primary VOCs, except phenols and a dihydrofuran, which

can be intermediates of the atmospheric oxidation of (alkyl)benzenes (Atkinson and Arey, 2003) and
long-chain alkanes (Aimanant and Ziemann, 2013; Strollo and Ziemann, 2013; Ranney and Ziemann,
2016), respectively. Nevertheless, only the phenol production may occur in high-NO OFRs, as the
particle-phase reaction in the photochemical formation of dihydrofurans from alkanes is too slow
compared to typical OFR residence times (Ranney and Ziemann, 2016). Therefore, the impact of $NO_3$
oxidation on VOC fate needs to be considered only if the OFR input flow contains high NO mixed with
biogenics and/or aromatics [(alkyl)benzenes and/or phenols]. However, (alkyl)benzenes were likely to
be major SOA precursors in, to our knowledge, the only few literature OFR studies with high NO levels
(Ortega et al., 2013; Tkacik et al., 2014; Liu et al., 2015). In the study of the air in a traffic tunnel (OFR185-
iNO mode; Tkacik et al., 2014), where toluene is usually a major anthropogenic SOA precursor as in
other urban environments (Dzepina et al., 2009; Borbon et al., 2013; Hayes et al., 2015; Jathar et al.,
2015), $NO_x$ was several hundreds of ppb. This resulted in an estimated $NO_{3exp}/OH_{exp}$ range of $\sim$0.1–1,
where up to $\sim$30% of cresols (intermediates of toluene oxidation) may have been consumed by $NO_3$.
Dihydrofurans may also have formed in the tunnel air (but outside the OFR) in the presence of $NO_x$
(Aimanant and Ziemann, 2013; Strollo and Ziemann, 2013) and, after entering the OFR, they would have
been substantially (up to $\sim$50%) consumed by $NO_3$. In the laboratory experiment of Liu et al. (2015) with
toluene, the injection of as much as 10 ppm NO elevated $NO_{3exp}/OH_{exp}$ to $\sim$100, where cresols from
toluene oxidation reacted almost exclusively with $NO_3$ in addition to being photolyzed.
**3.4.3 A case study**

We use a case study of an OFR254-13-iNO laboratory experiment with a large amount of toluene

(5 ppm) and $NO^{in}$ (10 ppm) to illustrate how very high VOC and NO concentrations cause multiple types
of atmospherically irrelevant reactions in OFR. Due to very high $OHR_{ext}$ and $NO^{in}$, photolysis of toluene
at 254 nm may have been important (Peng et al., 2016). In case of a high (close to 1) quantum yield, up
to $\sim$80% of the consumed toluene in their experiments could have been photolyzed (Scheme 1). Of the
rest of reacted toluene, $\sim$10% undergoes H-abstraction by OH from the methyl group in the model,
leading to an $RO_2$ similar to alkyl $RO_2$ and likely proceeding with normal $RO_2$ chemistry. $\sim$90% of the
toluene formed an OH-adduct (Calvert et al., 2002). As discussed above, 70% of this adduct (depending
on $NO^{in}$) is predicted to recombine with $NO_2$ producing nitroaromatics because of the ppm-level $NO_x$.
The adduct could also react with $O_2$ via two types of pathways, of which one was addition forming a
special category of $RO_2$ (OH-toluene-$O_2$ adducts) potentially undergoing ring-opening (Atkinson and
Arey, 2003; Orlando and Tyndall, 2012; Ziemann and Atkinson, 2012), the other H-elimination by $O_2$
producing cresols. Again, like toluene, cresols may have been substantially photolyzed. As a result of
$NO_{3exp}/OH_{exp}$ ~100, only a minor portion of cresol could have undergone OH addition and then H-
elimination again. This pathway leads to the formation of methyldihydroxybenzenes and other OH-
oxidation products (Atkinson and Arey, 2003). The rest of cresols may have formed methylphenoxy
radicals, nevertheless, dominantly via H-abstraction by $NO_3$, since H-abstraction by OH was even a minor
pathway compared to the OH-addition one (Atkinson et al., 1992). In summary, the model results
suggest that there were two possible routes leading to nitroaromatic formation. However, one of them
(recombination of OH-aromatic adducts with $NO_2$) is likely of little atmospheric relevance due to very
high $NO_x$ needed, and the other (H-abstraction from cresol) occurs in the atmosphere but is not a major
fate of aromatics (Calvert et al., 2002).
**3.5    Implications for OFR experiments with combustion emissions as input**
Emissions from combustion sources, e.g., vehicles and biomass burning, usually contain VOCs
and $NO_x$ at very high concentrations (Table 1). An injection of this type of emissions (typically with $OHR_{ext}$
of thousands of $s^{-1}$ or larger and $NO^{in}$ of tens of ppm or larger) in OFRs without any pretreatment is likely
to cause all experimental issues discussed in Peng et al. (2016) and this paper, i.e., strong OH
suppression, substantial non-tropospheric photolysis, strong $RO_2$ suppression by $NO_2$ whether $RO_2$ is
acyl $RO_2$ or not, fast reactions of $NO_2$ with OH-aromatic hydrocarbon adducts, substantial $NO_3$
contribution to VOC fate, and even a near-total inhibition of OFR chemistry due to complete titration of
$O_3$ by NO in the case of OFR254. We take the study of Karjalainen et al. (2016), who used an OFR to
oxidize diluted car exhaust in real-time, as an case study to investigate the extent to which these issues
may affect typical combustion source studies and to explore approaches to mitigate the problems.
During the first 200 s of their experiment (defined as the "cold start" period when the catalyst is
cold and emissions are high), NO and total hydrocarbon in the emissions of the test vehicle reached
~400 and ~600 ppm, respectively. We first simulate the oxidation of those emissions without any
dilution (even though x12 dilution was used in their experiments) to explore the most extreme
conditions. Our model simulation indicates that such an extremely concentrated source would generally
lead to bad high- or low-NO conditions (depending on NO concentration) in their OFR (Fig. 7), even
though it was run at relatively high $H_2O$ and UV. OH suppression can be as high as 3 orders of magnitude;
VOC fates by non-tropospheric photolysis and reactions of alkenes and phenols with $NO_3$ can be nearly
100%; up to ~1/3 of OH-toluene adduct may be recombined with $NO_2$ instead of forming an adduct with
$O_2$. After the test vehicle entered the "hot stabilized" stage (200–1000 s), its VOC emissions (on the
order of ppm) were still too high for an undiluted OFR to yield a good condition (Fig. S9). OH suppression
can still reach 2 orders of magnitude; non-tropospheric photolysis, and sometimes reactions with $NO_3$,
can still dominate over reactions with OH in VOC fates; reactions of OH-toluene adduct with $NO_2$ can
still be substantial at some small NO emission spikes. Moreover, although NO emissions were roughly
at ppm level even during the hot stabilized period, NO effective lifetime may be very short during that
period, leading to low-NO conditions in their OFR.
As suggested in Peng et al. (2016) for low-NO OFR, dilution of sources can also mitigate strong
deviations on OFR-iNO chemistry vs. atmospherically-relevant conditions. A dilution by a factor of 12,
as actually used by Karjalainen et al. (2016), appears to be sufficient to bring most of the hot stabilized
period under good conditions (Fig. S9). However, most VOC, or in other words, most SOA formation
potential, was emitted during the cold start period, when risky and bad conditions still prevailed (Figs.
7 and 8). Even if the emissions are diluted by x100, the cold-start emission peak (Fig. 7) is still under
risky conditions. Although bad conditions are eliminated and good condition is present during most of
time, this emission peak under risky condition may contribute >50% to total SOA formation potential
(Fig. 8). For SOA formed under good condition to be dominant, a dilution factor >400 would be needed.
Note that a strong dilution lowers aerosol mass loading in vehicle emissions. As a result, condensation
of gases onto particles is slower than in raw exhausts. However, condensational sinks after dilution may
still be significantly higher than typical ambient values (Matti Maricq, 2007; Donahue et al., 2016).
Note that the emissions of the test vehicle of Karjalainen et al. (2016) are rather clean compared
to the typical 2013 US on-road fleet (i.e., all at the hot stabilized stage) measured by Bishop and
Stedman (2013) (Figs. 9 and S10). For emissions of an average on-road fleet, a dilution by a factor of
100 or larger would be necessary to ensure that most emissions would be processed in OFR185 under
good conditions at the highest $H_2O$ and UV in this study (Figs. 9b and S10b,e,h). In the case of lower $H_2O$
and/or UV, an even larger dilution factor would be required.
Conducting OFR185-iNO experiments at high UV lowers the dilution factor needed for good
conditions. However, it also renders good high-NO condition impossible (see Section 3.2 and Fig. S4). If
one wants to oxidize vehicle exhausts in a high-NO environment in OFR, as in an urban atmosphere,
OFR185 at low UV is necessary. Consequently, a much stronger dilution is in turn necessary to keep the
operation condition still good. Nevertheless, not all vehicle emissions can be moved into good high-NO
region through a simple dilution (Figs. 9c and S10c,f,i). Furthermore, a low UV would seriously limit the
highest $OH_{exp}$ that OFR can achieve (~$3 \times 10^{11}$ molecules $cm^{-3}$ s for modeled good high-NO conditions in
this study), while a much higher $OH_{exp}$ would be desirable to fully convert SOA formation potential into
measurable SOA mass. If both good high-NO condition and high $OH_{exp}$ are required, new techniques
(e.g., injection of $N_2O$ at percent level proposed by Lambe et al. (2017)) may be necessary.
**4    Conclusions**
In this study, OFR chemistry involving $NO_y$ species was systematically investigated over a wide
range of conditions. NO initially injected into the OFR was found to be rapidly oxidized under most
conditions. In particular, due to high $O_3$ concentrations, NO lifetime in OFR254-iNO was too short to
result in a significant $RO_2$ consumption by NO compared to that by $HO_2$ under all conditions with active
chemistry. Nevertheless, it is not completely impossible for OFR185-iNO to have a significant $RO_2$ fate
by NO and minor non-tropospheric photolysis at the same time ("good high-NO conditions"). According
to our simulations, these conditions are most likely present at high $H_2O$, low UV, low $OHR_{ext}$, and $NO^{in}$
of tens to hundreds of ppb.
However, many past OFR studies with high NO injection were conducted under conditions
remarkably different from the abovementioned very narrow range. $NO^{in}$ and/or $OHR_{ext}$ in those studies
were often much higher than good high-NO conditions require (particularly, >3 orders of magnitude in
some OFR studies using combustion emissions as input). In addition to non-tropospheric organic
photolysis, OFR oxidation of highly concentrated sources can cause multiple large deviations from
tropospheric OH oxidation, i.e., $RO_2$ suppression by high $NO_2$, substantial nitroaromatic formation from
the recombination of $NO_2$ and OH-aromatic adducts, and fast reactions of VOCs with $NO_3$ compared to
those with OH.
Working at lower $NO_x$ (sub-ppm level) and VOC concentrations or dilution can mitigate these
experimental problems. In general, a strong dilution (by a factor of >100) is needed for OFR that process
typical on-road vehicle emissions. Humidification can also make good conditions more likely. By these
measures, good conditions can be guaranteed, as long as NO and/or precursor concentrations are
sufficiently low, while high-NO conditions cannot be ensured. To aid design and interpretation of OFR
experiments with high NO injection, we provide our detailed modeling results in a visualized form (Fig.
S3). For OFR users in need for both high $OH_{exp}$ and high NO, simple NO injection is not a good option.
New techniques (e.g., injection of $N_2O$ proposed by Lambe et al. (2017) or other innovations) may be
necessary to meet this need.


**Acknowledgements**
This work was partially supported by DOE (BER/ASR) DE-SC0011105 & DE-SC0016559, EPA STAR
83587701-0, and NSF AGS-1360834. We thank Pengfei Liu, Andrew Lambe, and Daniel Tkacik for
providing some OFR experimental data, the authors of Karjalainen et al. (2016) and their project IEA-
AMF Annex 44 for providing the data and information for the vehicle tests, Gary Bishop for providing
on-road vehicle emission data, and Andrew Lambe and William Brune for useful discussions.

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

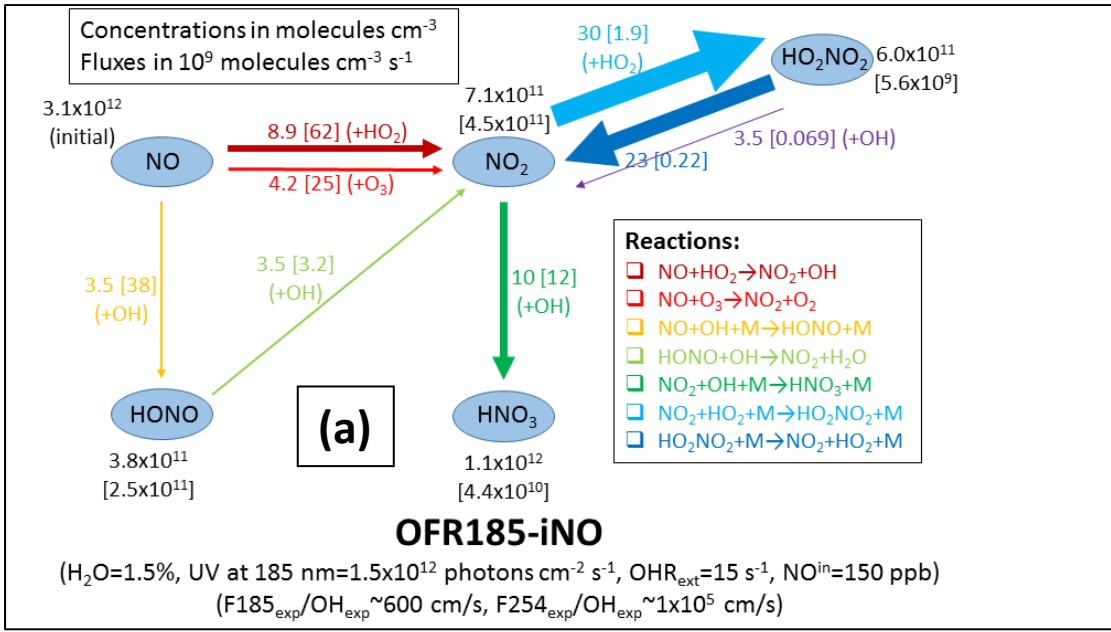

**(a) OFR185-iNO**

Concentrations in molecules cm$^{-3}$
Fluxes in $10^9$ molecules cm$^{-3}$ s$^{-1}$

NO $3.1\times10^{12}$ (initial)

$8.9\ [62]\ (+HO_2)$
$4.2\ [25]\ (+O_3)$

NO$_2$ $7.1\times10^{11}$ $[4.5\times10^{11}]$

$30\ [1.9]\ (+HO_2)$
$23\ [0.22]$

HO$_2$NO$_2$ $6.0\times10^{11}$ $[5.6\times10^9]$

$3.5\ [0.069]\ (+OH)$

$3.5\ [38]\ (+OH)$
$3.5\ [3.2]\ (+OH)$
$10\ [12]\ (+OH)$

HONO $3.8\times10^{11}$ $[2.5\times10^{11}]$

HNO$_3$ $1.1\times10^{12}$ $[4.4\times10^{10}]$

Reactions:
- $NO+HO_2\rightarrow NO_2+OH$
- $NO+O_3\rightarrow NO_2+O_2$
- $NO+OH+M\rightarrow HONO+M$
- $HONO+OH\rightarrow NO_2+H_2O$
- $NO_2+OH+M\rightarrow HNO_3+M$
- $NO_2+HO_2+M\rightarrow HO_2NO_2+M$
- $HO_2NO_2+M\rightarrow NO_2+HO_2+M$

(H$_2$O=1.5%, UV at 185 nm=$1.5\times10^{12}$ photons cm$^{-2}$ s$^{-1}$, OHR$_{ext}$=15 s$^{-1}$, NO$^{in}$=150 ppb)
(F185$_{exp}$/OH$_{exp}$~600 cm/s, F254$_{exp}$/OH$_{exp}$~$1\times10^5$ cm/s)


**(b) OFR185-iNO**

Concentrations in molecules cm$^{-3}$
Fluxes in $10^9$ molecules cm$^{-3}$ s$^{-1}$

NO $6.2\times10^{14}$

$160\ (+HO_2)$
$900\ (+O_3)$

NO$_2$ $9.1\times10^{13}$

$4.1\ (+HO_2)$
$3.0$

HO$_2$NO$_2$ $7.6\times10^{10}$

$0.29\ (+O_3)$

NO$_3$ $1.1\times10^8$

$210\ (+OH)$
$6.2\ (+OH)$
$57\ (+OH)$

HONO $2.1\times10^{13}$

HNO$_3$ $3.6\times10^{12}$

$15\ (+NO_3)$
$9.9$
$15\ (+NO_2)$

N$_2$O$_5$ $2.9\times10^{11}$

Reactions:
- $NO+HO_2\rightarrow NO_2+OH$
- $NO+O_3\rightarrow NO_2+O_2$
- $NO+OH+M\rightarrow HONO+M$
- $HONO+OH\rightarrow NO_2+H_2O$
- $NO_2+OH+M\rightarrow HNO_3+M$
- $NO_2+HO_2+M\rightarrow HO_2NO_2+M$
- $HO_2NO_2+M\rightarrow NO_2+HO_2+M$
- $NO_2+O_3\rightarrow NO_3+O_2$
- $NO_2+NO_3+M\rightarrow N_2O_5+M$
- $N_2O_5+M\rightarrow NO_2+NO_3+M$

(H$_2$O=1%, UV at 185 nm=$1\times10^{13}$ photons cm$^{-2}$ s$^{-1}$, OHR$_{ext}$=3000 s$^{-1}$, NO$^{in}$=30 ppm, T=295 K)
(F185$_{exp}$/OH$_{exp}$~$2\times10^5$ cm/s, F254$_{exp}$/OH$_{exp}$~$2\times10^7$ cm/s)


**(c) OFR254-7-iNO**

Concentrations in molecules cm$^{-3}$
Fluxes in $10^9$ molecules cm$^{-3}$ s$^{-1}$

NO $3.1\times10^{12}$ (initial)

$84\ [8300]\ (+O_3)$

NO$_2$ $6.3\times10^{11}$ $[1.4\times10^{12}]$

$22\ [15]\ (+HO_2)$
$16\ [0.62]$

HO$_2$NO$_2$ $4.2\times10^{11}$ $[1.6\times10^{10}]$

$4.8\ [0.11]\ (+OH)$
$2.5\ [6.0]\ (+O_3)$
$2.3\ [0.060]\ (+HO_2)$

NO$_3$ $1.4\times10^{10}$ $[1.8\times10^9]$

$13\ [15]\ (+OH)$
$7.5\ [6.2]\ (+NO_3)$
$6.7\ [0.25]$
$7.5\ [6.2]\ (+NO_2)$

HNO$_3$ $1.6\times10^{12}$ $[2.2\times10^{10}]$

N$_2$O$_5$ $2.0\times10^{11}$ $[7.2\times10^9]$

Reactions:
- $NO+O_3\rightarrow NO_2+O_2$
- $NO_2+OH+M\rightarrow HNO_3+M$
- $NO_2+HO_2+M\rightarrow HO_2NO_2+M$
- $HO_2NO_2+M\rightarrow NO_2+HO_2+M$
- $HO_2NO_2+OH\rightarrow NO_2+H_2O+O_2$
- $NO_3+HO_2\rightarrow NO_2+OH+O_2$
- $NO_2+O_3\rightarrow NO_3+O_2$
- $NO_2+NO_3+M\rightarrow N_2O_5+M$
- $N_2O_5+M\rightarrow NO_2+NO_3+M$

(H$_2$O=1.5%, UV at 254 nm=$3.4\times10^{14}$ photons cm$^{-2}$ s$^{-1}$, OHR$_{ext}$=15 s$^{-1}$, NO$^{in}$=150 ppb, T=295 K)


864 **Figure 1.** Schematics of main N-containing species and their major interconversion pathways under
865 typical input conditions for (a) OFR185-iNO with $NO^{in}$=150 ppb, (b) OFR254-7-iNO with $NO^{in}$=150 ppb,
866 and (c) OFR185-iNO with $NO^{in}$=30 ppm. Species average concentrations (in molecules $cm^{-3}$) are shown
867 in black beside species names. Arrows denote directions of the conversions. Average reaction fluxes (in
868 units of $10^9$ molecules $cm^{-3}$ $s^{-1}$) are calculated according to the production rate, and shown on or beside
869 the corresponding arrows and in the same color. Within each schematic, the thickness of the arrows is
870 a measure of their corresponding species flux. Multiple arrows in the same color and pointing to the
871 same species should be counted only once for reaction flux on a species. Note that all values in these
872 schematics are average ones over the residence time, except for those in square brackets in panels a
873 and b, which are average values within approximate NO effective lifetime ($\tau_{NO}$, or more accurately, an
874 integer multiple of the model's output time step closest to NO effective lifetime). All concentrations and
875 fluxes have two significant digits.

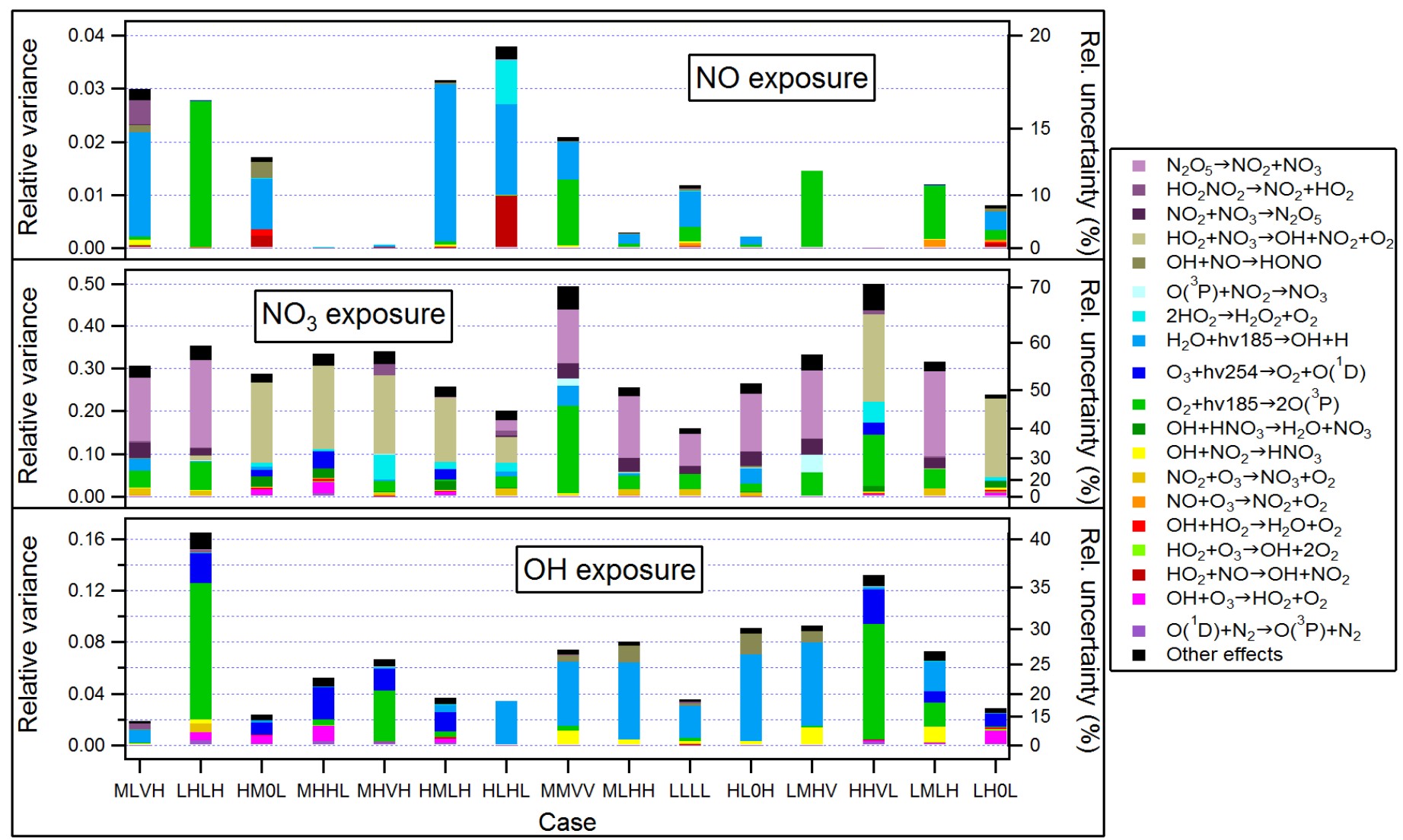

**Figure 2.** Relative variances (left axes)/uncertainties (right axes) of several outputs (i.e., NO, $NO_3$, and OH exposures) of Monte Carlo uncertainty propagation, and relative
contributions of key reactions to these relative variances in several typical cases (denoted in 4-character labels, see Table 2 for the typical case label code) in OFR185-iNO.
Relative variances are shown in linear scales (left axis), while corresponding relative uncertainties, equal to relative variances' square roots, are indicated by the non-linear
right axis. Only the reactions with a contribution of no less than 0.04 to at least one relative variance are shown.

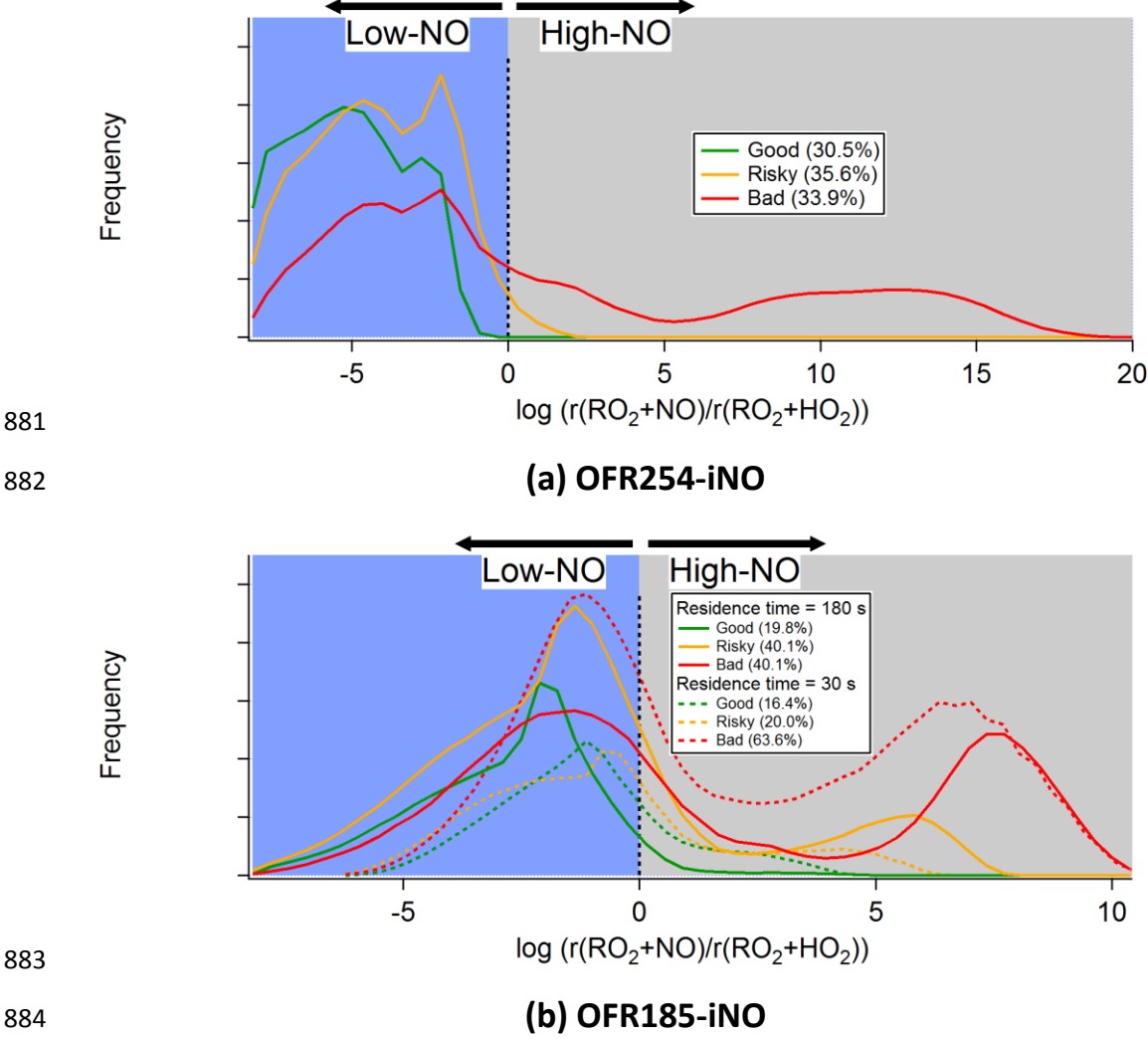



**(a) OFR254-iNO**



**(b) OFR185-iNO**

**Figure 3.** Frequency occurrence distributions of good, risky, and bad conditions (see Table 3) over
logarithm of the ratio between $RO_2$ reacted with NO and with $HO_2$ (see Section S1 for more detail) for
(a) OFR254-iNO (only the case with a residence time of 180 s) and (b) OFR185-iNO (including two cases
with residence times of 180 and 30 s). Low and high-NO regions (see Table 3) are colored in light blue
and grey, respectively.

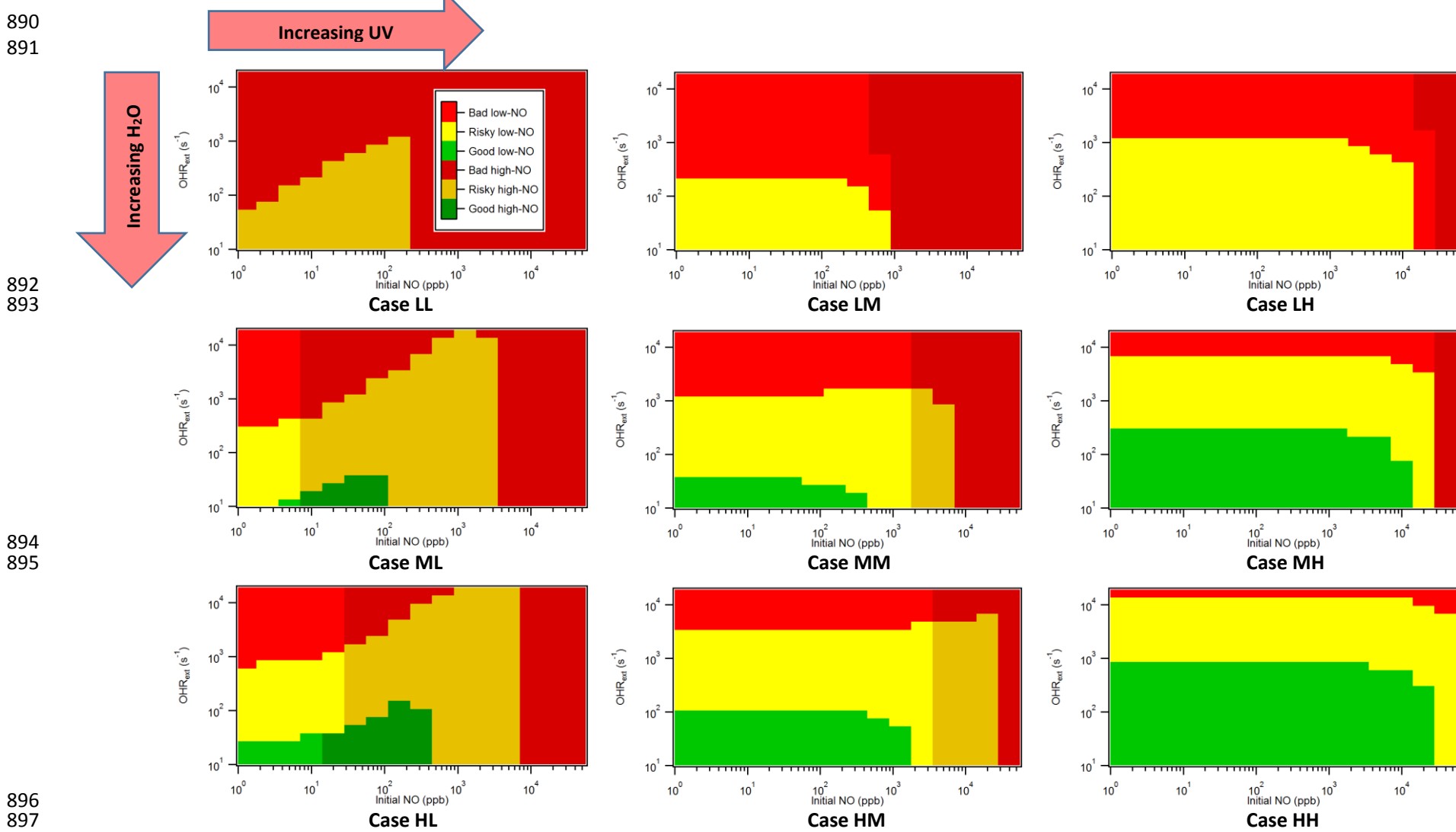

**Figure 4.** Image plots of the condition types defined in Table 3 vs. external OH reactivity (excluding N-containing species) and initial NO for several typical cases in OFR185-
iNO (see Table 2 for the case label code).

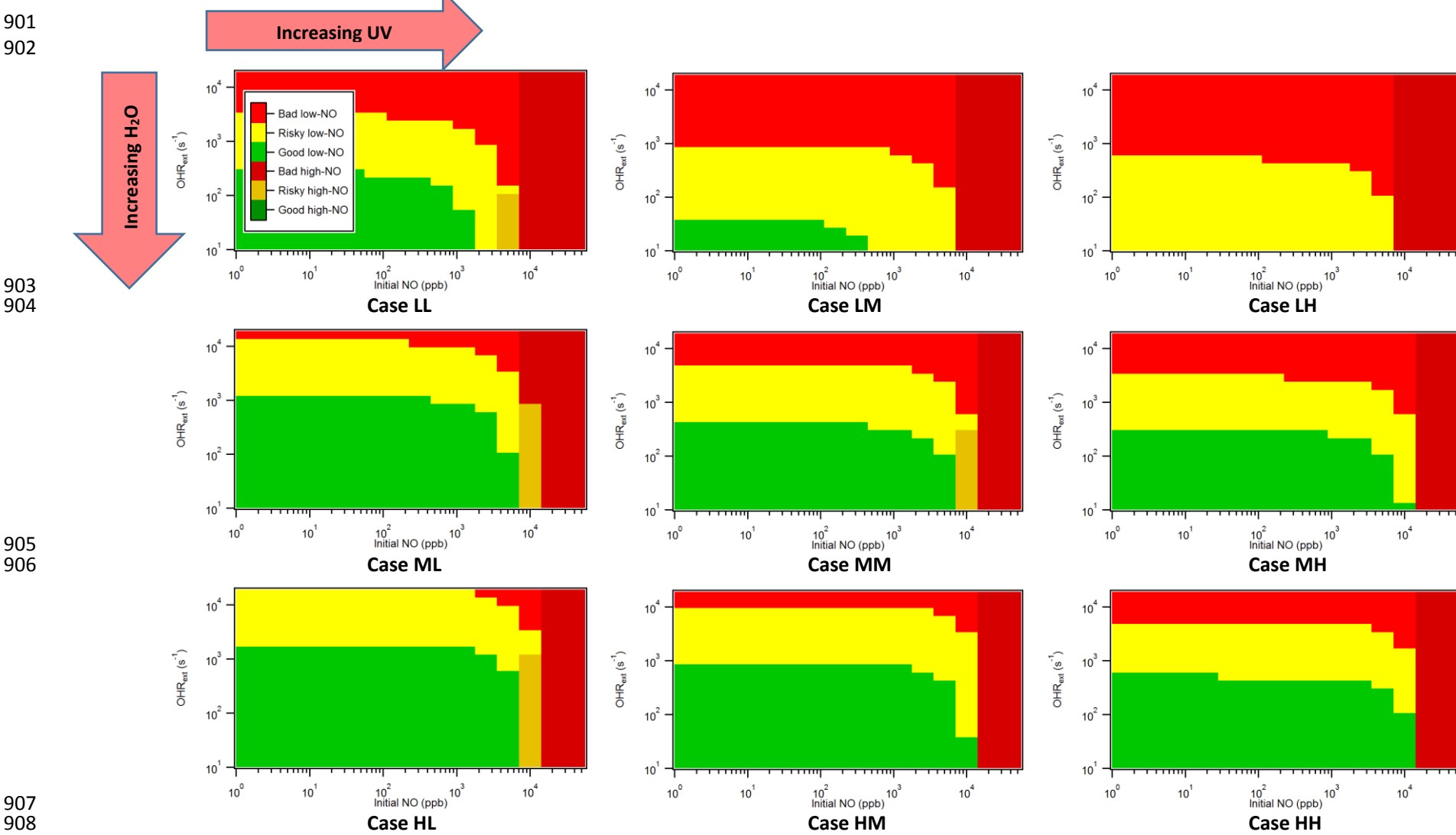


Increasing UV

Increasing H$_2$O

Case LL

Case LM

Case LH

Case ML

Case MM

Case MH

Case HL

Case HM

Case HH




**Figure 5.** Same format as Fig. 4, but for OFR254-22-iNO.

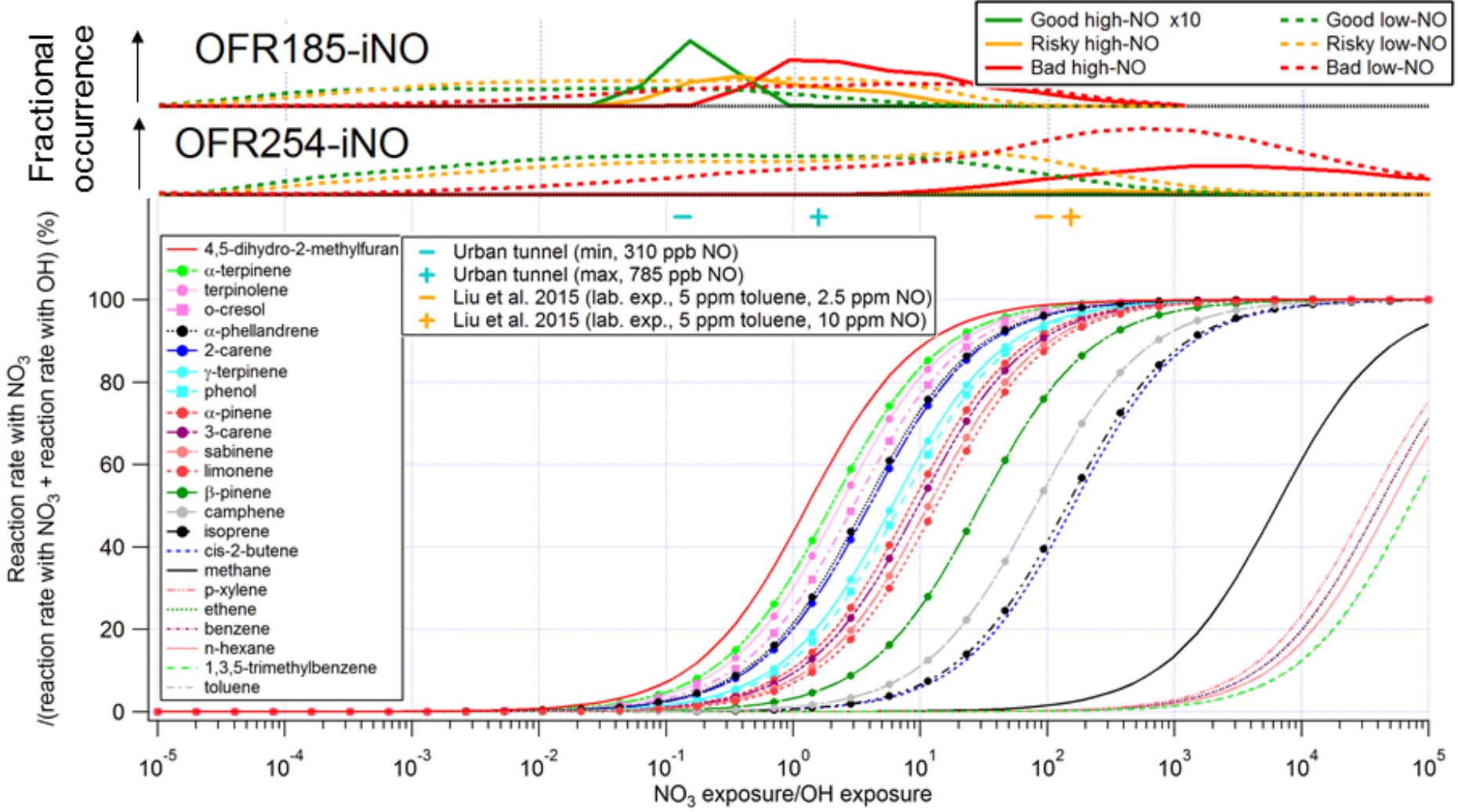

**Figure 6.** Fractional importance of the reaction rate of several species of interest with NO₃ vs. that with OH, as a function of the ratio of exposure to NO₃ and OH. The curves of biogenics and phenols are highlighted by solid dots and squares, respectively. The turquoise and orange markers show the ranges of modeled exposure ratios between NO₃


and OH of a source study in an urban tunnel (Tkacik et al., 2014) and a laboratory study (Liu et al., 2015) using OFR, respectively. In the upper part of the figure, the modeled
frequency distributions of ratios of $NO_3$ exposure to OH exposure under good/risky/bad high/low-NO conditions for OFR185-iNO and OFR254-iNO are also shown. See Table
3 for the definitions of the three types of conditions. All curves, markers, and histograms in this figure share the same abscissa.

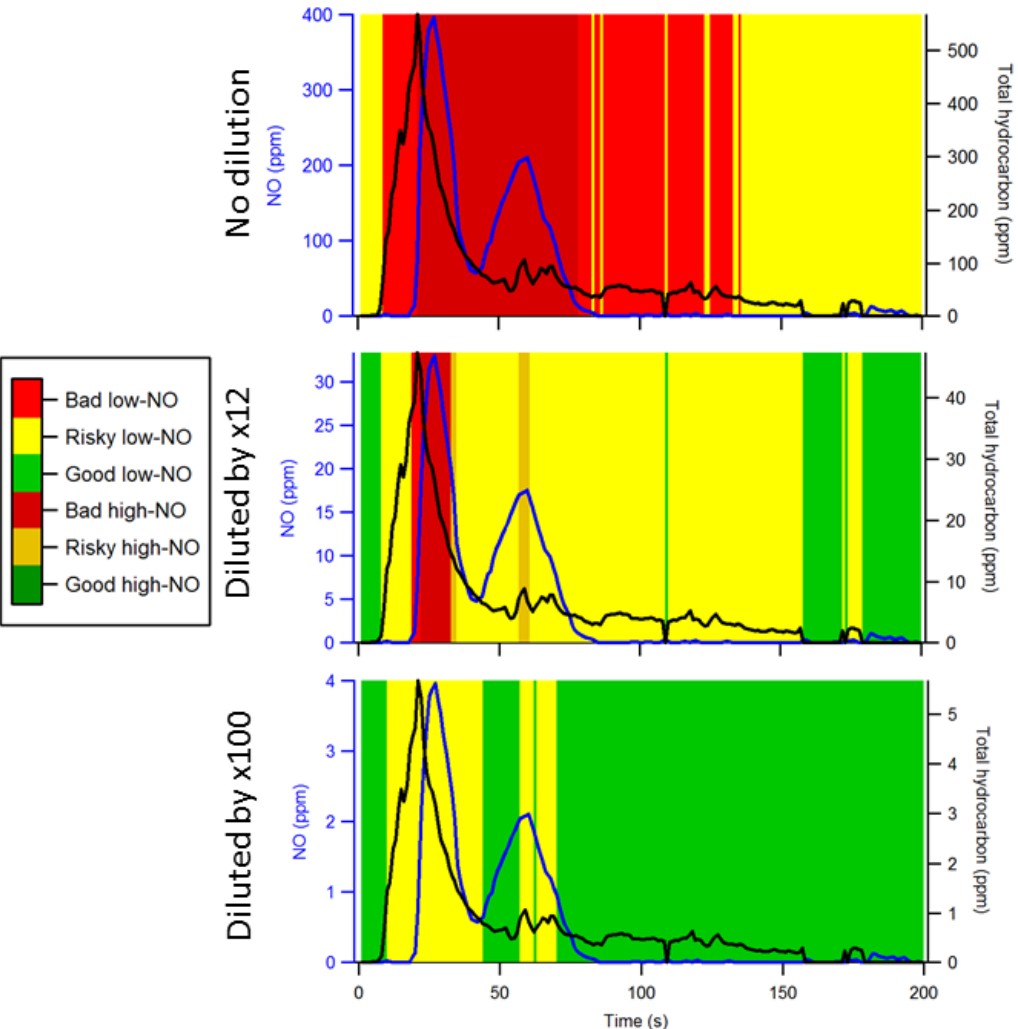

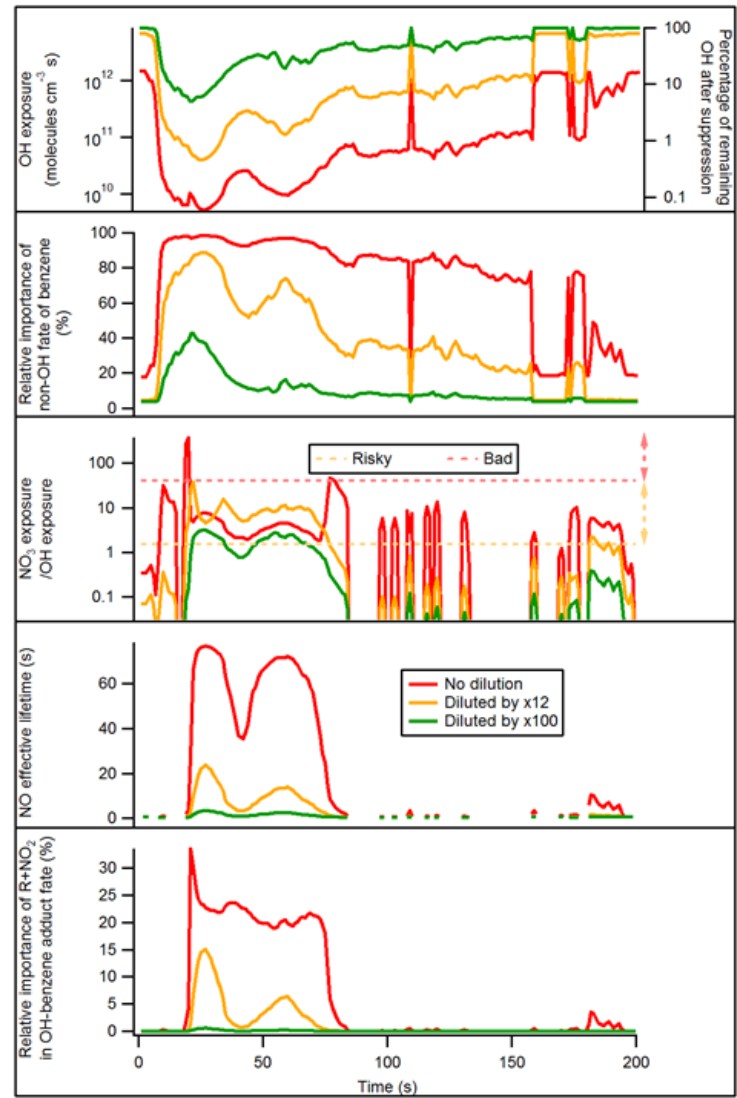


**Figure 7.** (left) NO and total hydrocarbon during the first 200 s of the test of Karjalainen et al. (2016) in the cases of no dilution, dilution by a factor of 12 (as actually done in that study), and dilution by a factor of 100. Different periods of time are colored according to corresponding emissions (i.e., input conditions for OFR), classified as good/risky/bad high/low-NO. (right) OH exposure/percentage of remaining OH after suppression, relative importance of non-OH fate of benzene, exposure ratio of $NO_3$ to OH, NO effective lifetime, and relative importance of reaction of OH-toluene adduct with $NO_2$ in the fate of this adduct in the OFR of Karjalainen et al. (2016) during the first 200 s of their test in the cases of no dilution, dilution by a factor of 12, and dilution by a factor of 100. Horizontal orange and red dashed lines in the middle right panel denote "risky" and "bad" regions for exposure ratio of $NO_3$ to OH, respectively. Above the orange (red) dashed line, reaction with $NO_3$ contributes >20% to the fate of phenol (isoprene).

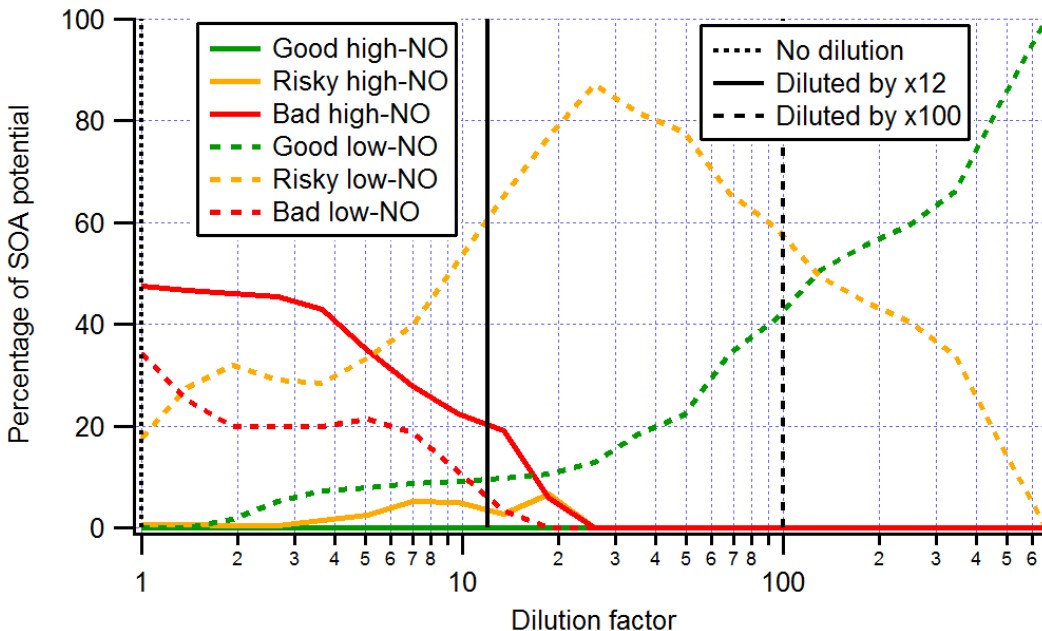

**Figure 8.** Secondary organic aerosol (SOA) potential (estimated from the total hydrocarbon
measurement) in the OFR of Karjalainen et al. (2016) formed during periods of time in the OFR
corresponding to good/risky/bad high/low-NO conditions, as a function of dilution factor. Vertical lines
denoting dilution factors of 1, 12 (as actually used in that study), and 100 are also shown.

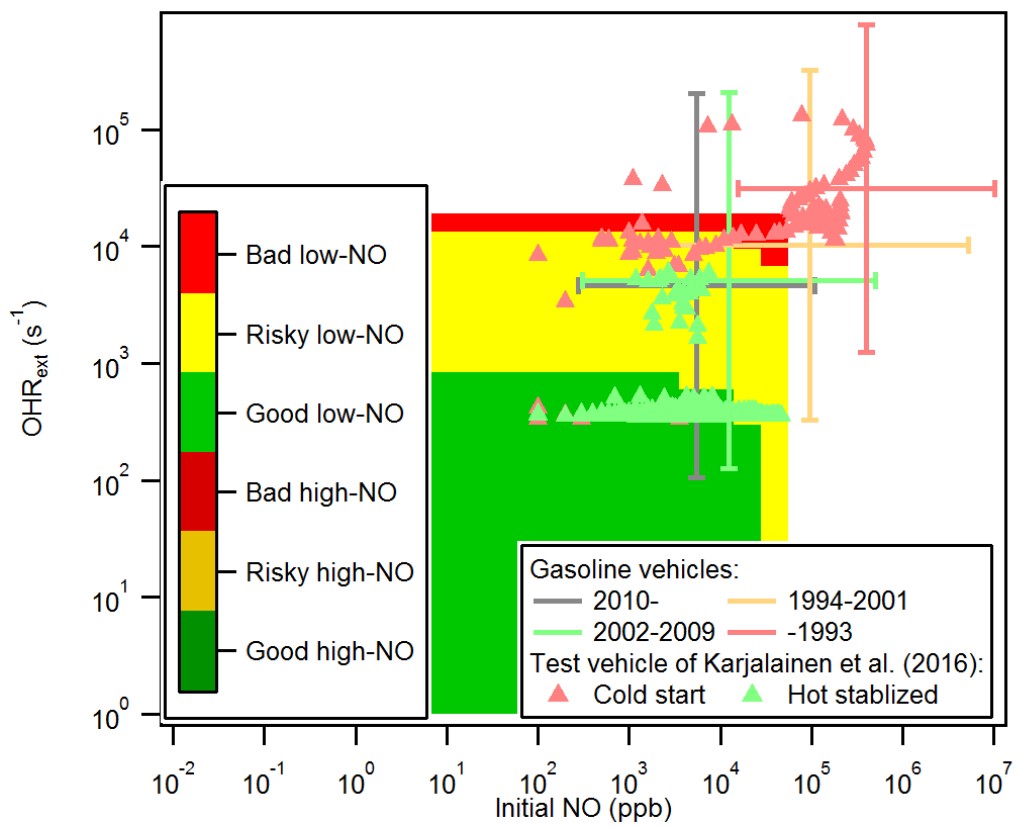


**(a) No dilution (background: Case HH)**

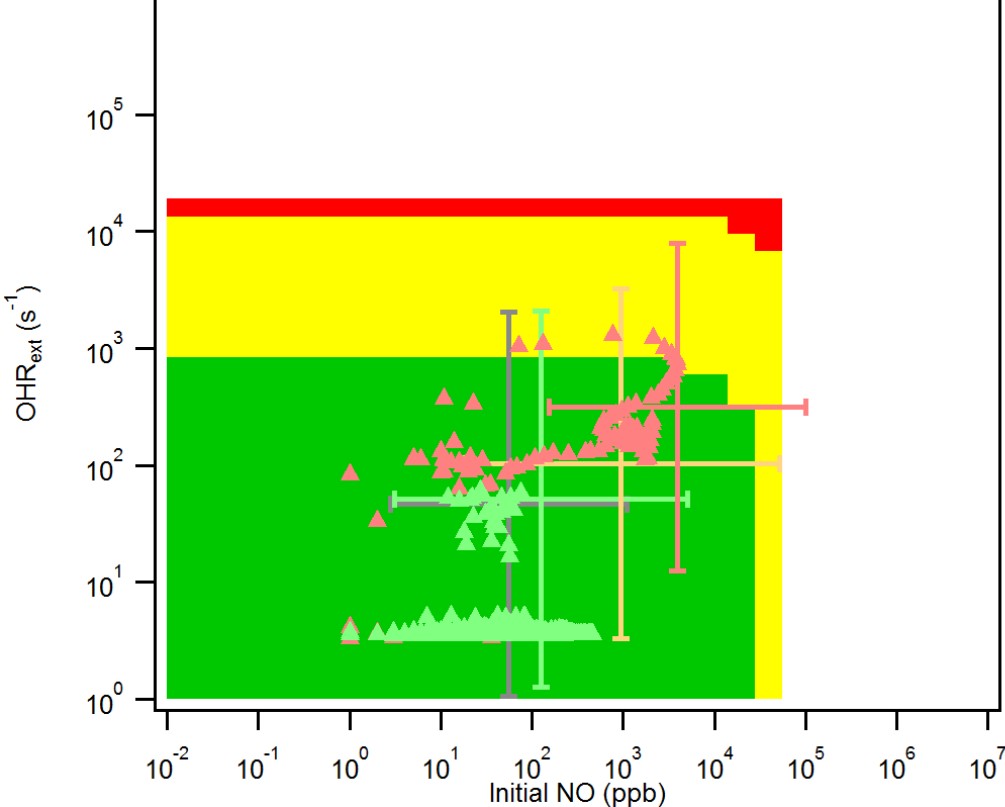


**(b) Dilution by a factor of 100 (background: Case HH)**

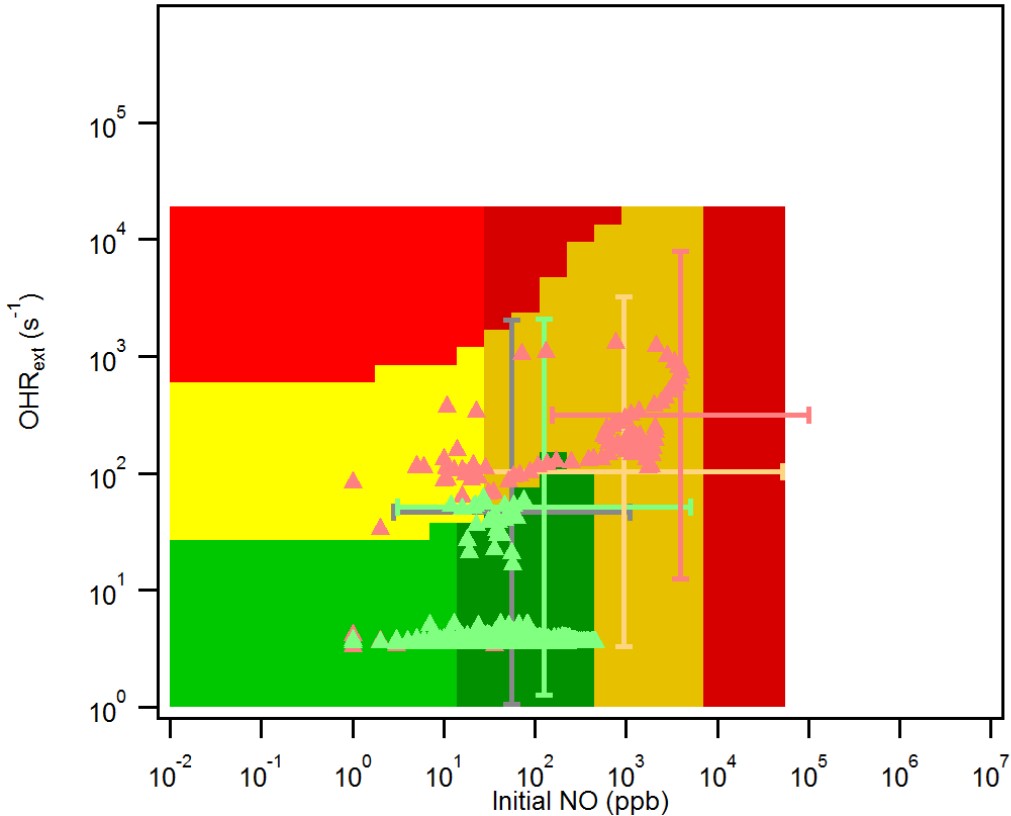


## (c) Dilution by a factor of 100 (background: Case HL)

**Figure 9.** Location of individual 1 s datapoints vs. OFR185-iNO reaction conditions. Datapoints are shown from the test vehicle of Karjalainen et al. (2016), as well as average exhaust from gasoline vehicle on-road emissions measured by Bishop and Stedman (2013). On-road emissions are classified by vehicle year and the distribution of each category is shown as a cross representing 1 standard deviation (with log-normal distribution assumed). The X and Y axes are NO and external OH reactivity (excluding N-containing species) due to vehicle emissions in OFR in the cases of (a) no dilution and (b,c) dilution by a factor of 100. The Karjalainen et al. (2016) points are classified as cold start (during first 200 s) and hot stabilized (during 200–1000 s). In addition, the same image plots as the panels of Cases HH (high $H_2O$ and high UV, see Table 2 for the case label code) and HL in Fig. 4 (OFR185-iNO) are shown as background for comparison.

**Scheme 1.** Possible major reactions in an OFR254-13-iNO with 5 ppm toluene and 10 ppm initial NO. Branching ratios in red are estimated by the model and/or according to
Calvert et al. (2002), Atkinson and Arey (2003), Ziemann and Atkinson (2012), and Peng et al. (2016). Note that addition/substitution on the aromatic ring may occur at other
positions. Intermediates/products shown here are the isomers that are most likely to form. Branching ratios shown in red are not overall but from immediate reactant.

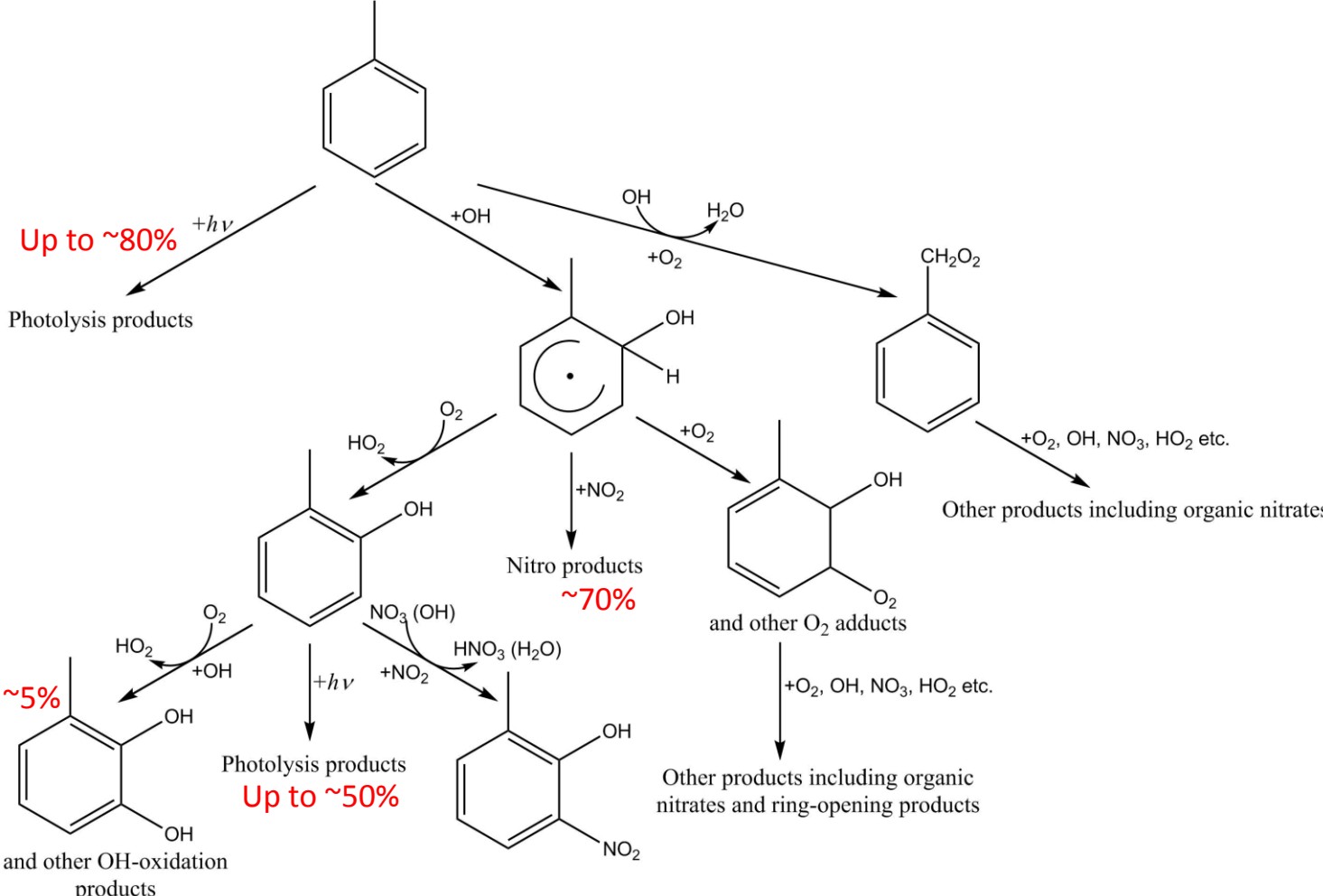


**Table 1.** Experimental conditions of several OFR studies with high NO injection.

| Study | Source type | Temperature (K) | Relative humidity (%) | Dilution factor | External OH reactivity of undiluted source ($s^{-1}$) | Source $NO_x$ concentration (ppm) |
|---|---|---|---|---|---|---|
| Link et al. (2016) | Diesel vehicle emission | | 50 | 45–110 | ~5000[*1] | 436[*1] |
| Martinsson et al. (2015) | Biomass burning emission | | | 1700 | 156400[*1] | 154 |
| Karjalainen et al. (2016) | Gasoline vehicle emission | 295 | 60 | 12 | ~73000[*2,a] | ~400[*1,b] |
| Liu et al. (2015) | Purified gas | 293 | 13 | 1 | ~1400[*1,a] | 10[*1,b] |
| Tkacik et al. (2014) | Tunnel air | 293 | 42 | 1 | ~60[*1,a] | ~0.8[*1] |
| Ortega et al. (2013) | Biomass burning emission | 290 | 30 | ~500 | ~15-500 | ~0.2 |

[*1] maximum value in the study
[*2] value at the moment of maximum NO emission
[a] $NO_y$ species excluded
[b] NO only

**Table 2.** Code of the labels of typical cases. A case label can be composed of four characters denoting the water mixing ratio, the photon flux, the external OH reactivity
excluding N-containing species, and the initial NO mixing ratio, respectively. A case label can also be composed of two characters denoting the water mixing ratio and the
photon flux.

|  | Water mixing ratio | Photon flux | External OH reactivity (no ON) | Initial NO mixing ratio |
|---|---|---|---|---|
| Options | L=low (0.07%) | L=low ($10^{11}$ photons cm$^{-2}$ s$^{-1}$ at 185 nm; $4.2 \times 10^{13}$ photons cm$^{-2}$ s$^{-1}$ at 254 nm) | 0 | 0 |
|  | M=medium (1%) | M=medium ($10^{13}$ photons cm$^{-2}$ s$^{-1}$ at 185 nm; $1.4 \times 10^{15}$ photons cm$^{-2}$ s$^{-1}$ at 254 nm) | L=low (10 s$^{-1}$) | L=low (10 ppb) |
|  | H=high (2.3%) | H=high ($10^{14}$ photons cm$^{-2}$ s$^{-1}$ at 185 nm; $8.5 \times 10^{15}$ photons cm$^{-2}$ s$^{-1}$ at 254 nm) | H=high (100 s$^{-1}$) | H=high (316 ppb) |
|  |  |  | V=very high (1000 s$^{-1}$) | V=very high (10 ppm) |
| Example | LH0V: | low water mixing ratio, high photon flux, no external OH reactivity (excluding ON), very high initial NO mixing ratio | | |
|  | ML: | medium water mixing ratio, low photon flux | | |


**Table 3.** Definition of condition types in this study (good/risky/bad high/low-NO).

| Condition | Good | Risky | Bad |
|---|---|---|---|
| Criterion | $F185_{exp}/OH_{exp}<3\times10^3$ cm s$^{-1}$ and $F254_{exp}/OH_{exp}<4\times10^5$ cm s$^{-1}$ | $F185_{exp}/OH_{exp}<1\times10^5$ cm s$^{-1}$ and $F254_{exp}/OH_{exp}<1\times10^7$ cm s$^{-1}$ (excluding good conditions) | $F185_{exp}/OH_{exp}\geq1\times10^5$ cm s$^{-1}$ or $F254_{exp}/OH_{exp}\geq1\times10^7$ cm s$^{-1}$ |
| **Condition** | **High-NO** | | **Low-NO** |
| Criterion[*] | $\dfrac{r(RO_2+NO)}{r(RO_2+HO_2)}>1$ | | $\dfrac{r(RO_2+NO)}{r(RO_2+HO_2)}\leq1$ |

[*] See Section S1 for detail.

**Table 4.** Statistics of the ratio between OH exposures calculated in the model with the Lambe et al. (2011) residence time distribution ($OH_{exp,RTD}$) and in the plug-flow model
($OH_{exp,PF}$). The geometric mean, uncertainty factor (geometric standard deviation), and percentage of outlier cases (>3 or <1/3) are shown for OFR185-iNO, OFR254-70-iNO,
and OFR254-7-iNO.

|  | Geometric mean | Uncertainty factor | Outlier cases (%) |
|---|---|---|---|
| OFR185-iNO | 1.91 | 1.64 | 11 |
| OFR254-7-iNO | 1.59 | 1.51 | 7 |
| OFR254-70-iNO | 1.48 | 1.29 | 3 |
