# Peer review of "Modeling of the chemistry in oxidation flow reactors with high initial NO"

_Atmospheric Chemistry and Physics, 2017_

## Referee Comment (RC1) · Anonymous Referee #1 · 8 Jun 2017

In the study at hand, a previously developed kinetic box model is applied to a plethora of scenarios that could be encountered when using oxidation flow reactors (OFR) to produce secondary organic aerosol (SOA) in the presence of NO. Peng et al. present a very detailed study that, while not directly relevant for the general public, might be very helpful for the specialized field of atmospheric researchers employing OFR and falls within the scope of ACP. Especially the comprehensive Fig. S7 should be a fantastic resource for research groups working with OFR and without excess to kinetic modelling tools. The authors convincingly show that the conditions in which OFR are often operated are far from atmospheric relevance. The article is well-structured, but is now and then difficult to read, e.g. in Sects. 3.1.1 and 3.2. A reason for this might be that the narrative doesn't closely follow the figures, and, while the figures contain lots of useful information, it seems that much of the given information is not discussed in the manuscript, which would technically render most of the figures in the main text supplementary material. I would like to encourage the authors at this point to re-think their use of figures in this manuscript. For example, can the information in Figure 1 be presented in a more concise, meaningful way? It also does not help that positions and sizes of fonts and symbols in Fig. 1 are different in all three panels. This does not diminish the solid scientific message of this work, but would help immensely to reach a larger audience. Thus, I can recommend this paper for publication in ACP after only minor revisions, but would encourage the authors to revise the visual presentation of their scientific results. Further point-by-point comments are given below.

**General Comments**

- The authors have to define "non-tropospheric" photolysis, which shows up as early as in the abstract, but is never properly defined. Is the connotation of stratospheric or mesospheric photolysis intended?
- Why have the authors chosen the term "risky" for conditions that are not unambiguously good or bad? What is the "risk" that is taken here? It would be helpful to briefly motivate the use of this word around l. 171.
- Fig. 2: What is shown on the x-axis? Please label/explain these cases. This is also relevant in the later discussion, around l. 323.
- l. 295: You compare NO lifetime to reactor residence time. Should it not be better to compare to e.g. VOC lifetime in the reactor, or generally to total overturn of reactive material? I can imagine a scenario where NO is used up very quickly, but so are all other reactive gases, so that much of the reactor residence time is not used to make (or age) SOA and hence mostly irrelevant anyhow.
- l. 299: Figure 3 is very complex, yet is doesn't find much introduction. Please expand your discussion of this figure the first time it is referenced in the text.

**Minor Comments**

- l. 40: "on similar timescales"
- l. 41: Is there an "of" missing after "decoupling"? Alternatively: "… to decouple …"
- l. 72: Please give a unit of exposure. Also relevant e.g. in line 197.
- l. 275: Instead of "similar with those cases" it must read "similar to those cases".
- l. 394: "Despite its double bond, ethene reacts as slowly with NO3 as alkanes, likely due to lack of alkyl groups enriching electron density on the C=C bond, which slows NO3 addition." Why is this relevant here?
- l. 439-441: This sentence is confusing, the word "process" seems out of place here.
- l. 444-447: In this sentence, please briefly state again (maybe in parenthesis) which route is which in this example (H-abstraction vs. OH addition) to avoid confusion.
- l. 465: "… suppression can as high …" should read "… suppression can be as high… ".

- l. 477: "most hot stabilized period". Is there a word missing here?
- Fig. S1: please use consistent colors for chemical species.

---

## Referee Comment (RC2) · Anonymous Referee #2 · 25 Jun 2017

The current paper explores a chemical space extended to consider high NO concentrations within an OFR. Such a contribution, whilst of limited interest outside the immediate field, should be of considerable worth to users of such devices, particularly those looking to explore the emissions from high temperature combustion sources. However, to some degree, the paper is missing the same point that many previous theoretical characterisations of the devices also miss. The chemical space is just one element influencing the performance and atmospheric relevance of all PAM-type reactors (and the gas phase chemical space, just a subset of this). It is for this reason that I would hope that the current study is envisaged as one of a series of papers that will be extended to the dynamical, physical and condensed-phase chemical considerations. I will return to this below.

Having said this, within the stated scope, this paper carries a large amount of good new work that will make it worthy of publication in ACP. The chemical modelling appears appropriate with most of the necessary rate constants relatively well-constrained. This allows the characterisation of "good", "risky" and "bad" conditions under both 185 and 254 nm photolysis, though with the same caveats to the earlier work relating to uncertainties in the photolysis cross-sections and product yields of all possible VOCs (particularly when considering complex mixtures as in combustion emissions). In general, I am in agreement with the other referee that the gas phase chemical modelling alone warrants publication in ACP, but would invite the authors to address two main points to establish the validity of the approach and one point relating to the contextualisation of their study.

i) Validity of the plug flow assumption: in section 3.1.3 it is stated that the uncertainties relating to kinetic parameters are relatively low compared to other factors including the plug flow assumption, referring to Peng et al., 2015. It would be useful for the authors to discuss whether the relative kinetic vs dynamic uncertainties under the high NOx conditions are of a comparable magnitude to those under low NOx conditions. There have been plenty of studies of axial and radial gradients in flow reactors, so some justification of the highly simplified modelling approach would be appropriate, given the biggest uncertainties are explicitly stated as being related to this assumption.

ii) Validity of separating the numerical treatment of gas phase and particle phase processes: there is no statement of any of the uncertainty in gas phase chemistry being attributable to multiphase processes. I find this rather curious, since the primary focus of most PAM chamber studies relates is particulate mass. Both radical and closed shell species may interact substantially with the particle phase. All the particulate material in SOA particles is, by definition, formed from the vapour phase. If the flow regime is anything near plug flow, then the particle number, condensation sink, mass and composition of the particulate will evolve with the gas phase species and hence mass transfer (in both directions, where there is oxidative fragmentation and functionalisation) will

be changing temporally and spatially inside the reactor. There really should be some discussion of the potential impacts of these processes in the paper.

iii) My final point relates to the context of the study. If it is not envisaged that this second paper on the chemical characterisation of OFRs is to eventually be accompanied by a numerical study of the multiphase processes, then I think the paper requires quite a bit more contextualisation. The root of the missing material relates to the competition between processes (nucleation, condensation, evaporation, coagulation, condensed phase reaction) alluded to in point ii) above and relating to aerosol dynamical evolution that are highly dependent on the magnitudes of different moments of the aerosol distribution. Extrapolation to concentration regimes other than the dilutions under the operating conditions of the OFR is simply not possible without the adoption of substantial questionable assumptions or use of a highly complex model which has yet to be described. The current paper implicitly aims to limit its scope to gas phase oxidation of VOCs in the OFR, but this is seldom the purpose to which they are put. Indeed, the limited context for OFR studies explicitly points to their use for "...secondary organic aerosol (SOA) formation and aging [studies], in both the laboratory and the field", because of the perceived advantage of elevated oxidant levels. None of the disadvantages that are related directly to the inappropriate exptrapolation of all the multiphase processes of relevance to SOA formation and transformation are mentioned. This requires significant rebalancing, ideally quantitatively in a further detailed publication but at least qualitatively in the introduction of the current paper.

Related to the above, the previous findings of nitrogen being incorporated into SOA are very tricky to interpret. The recommendations for operation are made from the perspective of gas phase oxidation to ensure that the gas phase product distribution is not anomalous. Incorporation of the nitrogenous species into particles will be subject to multiphase processes leading to net mass transfer between the phases. The mass transfer rate will be proportional to not only the difference between the gaseous concentration and the equilibrium concentration above the particle, but also to the condensation sink provided by the particles. Extrapolation to the amount of a component or class of components in the SOA (e.g. nitrogen-containing ones) to ambient conditions should not only consider the equivalent oxidant dose and gas phase chemical regime, but also aim to establish some equivalence in terms of the mass transfer between phases.

So, in summary, I would suggest that the material contained in the paper is readily publishable, but requires both contextualisation and more discussion of the likely uncertainties surrounding the assumed dynamical framework and separation from the condensed phase processes.

---

## Author Comment (AC1) · 15 Jul 2017

We thank the referees for their reviews. To facilitate the review process we have copied the reviewer comments in black text. Our responses are in regular blue font. We have responded to all the referee comments and made alterations to our paper (**in bold text**). Figures, tables, and sections in the responses are numbered as in the *revised* manuscript unless specified, while page and line numbers refer to the ACPD paper.

**Anonymous Referee #1**

In the study at hand, a previously developed kinetic box model is applied to a plethora of scenarios that could be encountered when using oxidation flow reactors (OFR) to produce secondary organic aerosol (SOA) in the presence of NO. Peng et al. present a very detailed study that, while not directly relevant for the general public, might be very helpful for the specialized field of atmospheric researchers employing OFR and falls within the scope of ACP. Especially the comprehensive Fig. S7 should be a fantastic resource for research groups working with OFR and without excess to kinetic modelling tools. The authors convincingly show that the conditions in which OFR are often operated are far from atmospheric relevance.

R1.0) The article is well-structured, but is now and then difficult to read, e.g. in Sects. 3.1.1 and 3.2. A reason for this might be that the narrative doesn't closely follow the figures, and, while the figures contain lots of useful information, it seems that much of the given information is not discussed in the manuscript, which would technically render most of the figures in the main text supplementary material. I would like to encourage the authors at this point to re-think their use of figures in this manuscript. For example, can the information in Figure 1 be presented in a more concise, meaningful way? It also does not help that positions and sizes of fonts and symbols in Fig. 1 are different in all three panels. This does not diminish the solid scientific message of this work, but would help immensely to reach a larger audience. Thus, I can recommend this paper for publication in ACP after only minor revisions, but would encourage the authors to revise the visual presentation of their scientific results. Further point-by-point comments are given below.

We have made the sizes of fonts and symbols identical in the 3 panels of Fig. 1. To improve the legibility of Sections 3.1.1 and 3.2, we have made modifications to the text in a number of places: reformulating/reordering sentences, adding/improving references to figures, clarifying some details etc. In particular, we have referred to Fig. S7 in the ACPD version (Fig. S3 in the revised version; other figures in SI also renumbered accordingly) in these sections as well as elsewhere in the paper to better take advantage of its large amount of useful information. Note that Fig. S7 (in the ACPD version) was included mainly for experimental planning purposes. We did not aim to explain every feature in Fig. S7 in the ACPD version and have only referred to it when useful, and have not substantially changed the text just according to the material in this figure.

The modified Sections 3.1.1 and 3.2 now read as follows:

Section 3.1:

[revised manuscript text omitted]

R1.1) The authors have to define "non-tropospheric" photolysis, which shows up as early as in the abstract, but is never properly defined. Is the connotation of stratospheric or mesospheric photolysis intended?

185 and 254 nm photons, the main driver of OH production in OFRs, do not exist in the troposphere. VOC photolysis at these wavelengths can only occur above the troposphere. We thus call it "non-tropospheric".

We have modified the following sentence in L79 to include a clarification of non-tropospheric photolysis:

**"Peng et al. (2016) systematically examined the relative importance of non-OH/non-tropospheric reactants on the fate of VOCs over a wide range of conditions, and provided guidelines for OFR operation to avoid non-tropospheric VOC photolysis, i.e., VOC photolysis at 185 and 254 nm."**

R1.2) Why have the authors chosen the term "risky" for conditions that are not unambiguously good or bad? What is the "risk" that is taken here? It would be helpful to briefly motivate the use of this word around l. 171.

We choose the word "risky" for conditions that are not unambiguously good or bad for experiments with *all* SOA precursors. Under risky conditions, some VOCs may have significant non-tropospheric photolysis while others may not. To further clarify the good/risky/bad conditions, we have modified the text in L172 to read:

**"Under good conditions, photolysis of most VOCs has a relative contribution <20% to their fate; under bad conditions, non-tropospheric photolysis is likely to be significant in all OFR experiments, since it can hardly be avoided for oxidation intermediates, even if the precursor(s) does not photolyze at all. Under risky conditions, some species photolyzing slowly and/or reacting with OH rapidly (e.g., alkanes, aldehydes, and most biogenics) still have a relative contribution of photolysis <20% to their fates, while species photolyzing more rapidly and/or reacting with OH more slowly (e.g., aromatics and other highly conjugated species and some saturated carbonyls) will undergo substantial non-tropospheric photolysis. Note that these definitions are slightly different than in Peng et al. (2016)."**

R1.3) Fig. 2: What is shown on the x-axis? Please label/explain these cases. This is also relevant in the later discussion, around l. 323.

We believe that the meaning of the typical case labels have been well defined in Table 2. For more clarity, we have modified the following text to better refer readers to that table.

In L150:

"**We explore physical input cases evenly spaced in a logarithmic scale over very wide ranges: $H_2O$ of 0.07%–2.3%, i.e., relative humidity (RH) of 2–71% at 295 K; 185 nm UV of $1.0x10^{11}$–$1.0x10^{14}$ and 254 nm UV of $4.2x10^{13}$–$8.5x10^{15}$ photons $cm^{-2}$ $s^{-1}$; $OHR_{ext}$ of 1–16000 $s^{-1}$; $O_{3,in}$ of 2.2–70 ppm for OFR254; initial NO mixing ratio ($NO^{in}$) from 10 ppt to 40 ppm. Besides, conditions with $OHR_{ext}$=0 are also explored. UV at 254 nm is estimated from that at 185 nm according to the relationship determined by Li et al. (2015). Several typical cases within this range as well as their corresponding 4 or 2-character labels (e.g., MM0V and HL) are defined in Table 2.**"

In L319:

"**In addition, a low $OHR_{ext}$ (generally <50 $s^{-1}$) and a higher $H_2O$ (the higher the better, although there is no apparent threshold) are also required for good high-NO conditions (Fig. S4), as Peng et al. (2016) pointed out. Risky high-NO conditions often occur between good and bad high-NO conditions, e.g., at lower $NO^{in}$ than bad conditions (e.g., Cases ML, MM, HL, and HM, see Table 2 for the typical case label code), at higher $OHR_{ext}$ and/or $NO_{in}$ than good conditions (e.g., Cases ML and MM), and at lower $H_2O$ than good conditions (e.g., Case LL).**"

In the caption of Fig. 2 (L756):

"**Figure 1. Relative variances (left axes)/uncertainties (right axes) of several outputs (i.e., NO, $NO_3$, and OH exposures) of Monte Carlo uncertainty propagation, and relative contributions of key reactions to these relative variances in several typical cases (denoted in 4-character labels, see Table 2 for the typical case label code) in OFR185-iNO.**"

R1.4) l. 295: You compare NO lifetime to reactor residence time. Should it not be better to compare to e.g. VOC lifetime in the reactor, or generally to total overturn of reactive material? I can imagine a scenario where NO is used up very quickly, but so are all other reactive gases, so that much of the reactor residence time is not used to make (or age) SOA and hence mostly irrelevant anyhow

We do not agree that a situation where "much of the reactor residence time is not used to make (or age) SOA and hence mostly irrelevant anyhow" after NO is used up quickly is likely. Even for

primary VOCs with lifetimes comparable with or shorter than that of NO, their oxidation intermediates/products actually have significant presence for a much longer period than NO lifetime (Nehr et al., 2014; Schwantes et al., 2017). Besides, heterogeneous OA oxidation can be important at high photochemical ages in OFR (Hu et al., 2016), leading to decomposition and revolatilization of particle-phase species. Whether $RO_2$ generated from these second and later generation species undergo high-NO or low-NO oxidation still matters in OFR chemistry, regardless of NO lifetime. Therefore, we believe that the entire residence time is the appropriate period of interest for the high/low-NO considerations.

In addition, we have investigated a case with much shorter residence time (30 s) to more focus on NO and primary VOC oxidation, as the further oxidation is limited by the short residence time. This case may be seen as closer to the Referee's scenario. However, the fraction of good high-NO conditions in this case is still comparable to that with a residence time of 180 s.

For more clarity, we have added the following sentence at the end of the first paragraph of Section S1 (L94 in SI):

"**The entire residence time is taken into account since there is still significant presence of VOCs after NO and primary VOCs are destroyed. The oxidation intermediates/products of primary VOCs can exist for a much longer period than NO lifetime (Nehr et al., 2014; Schwantes et al., 2017). In addition, heterogeneous OA oxidation can be important at high photochemical ages in OFR (Hu et al., 2016), leading to decomposition and revolatilization of particle-phase species. Thus continuing oxidation processes are very likely to occur during the entire the residence time.**"

R1.5) l. 299: Figure 3 is very complex, yet is doesn't find much introduction. Please expand your discussion of this figure the first time it is referenced in the text

To introduce Fig. 3, we modify the text to L298 to read:

"**Figure 3 shows the fractional occurrence distribution of good/risky/bad conditions in the entire explored condition space over logarithm of r($RO_2$+NO)/r($RO_2$+$HO_2$), which distinguishes high- and low-NO conditions. In OFR254-iNO, $\tau_{NO}$ is so short that no good high-NO condition is found in the explored range in this study (Fig. 3a).**"

R1.6) l. 40: "on similar timescales"

Corrected as the Referee suggested.

R1.7) l. 41: Is there an "of" missing after "decoupling"? Alternatively: "… to decouple …"

We have added "of" after "decoupling" and now the sentence reads:

"**Chemical reactors allow for decoupling of these two types of processes.**"

R1.8) l. 72: Please give a unit of exposure. Also relevant e.g. in line 197.

We have specified the units of all key quantities mentioned in this paragraph as well as in L197.

The modified text in L71 reads:

"**Li et al. (2015) and Peng et al. (2015) developed a box model for OFR HO$_x$ chemistry that predicts measurable quantities [e.g., OH exposure (OH$_{exp}$, in molecules cm$^{-3}$ s] and O$_3$ concentration (abbr. O$_3$ hereinafter, in ppm)] in good agreement with experiments. This model has been used to characterize HO$_x$ chemistry as a function of H$_2$O mixing ratio (abbr. H$_2$O hereinafter, unitless), UV light intensity (abbr. UV hereinafter, in photons cm$^{-2}$ s$^{-1}$), and external OH reactivity [in s$^{-1}$, OHR$_{ext}$=$\sum k_i c_i$, i.e., the sum of the products of concentrations of externally introduced OH-consuming species ($c_i$) and rate constants of their reactions with OH ($k_i$)].**"

And that around L197:

"**We evaluate this issue below by calculating NO effective lifetime ($\tau_{NO}$, in s), defined as NO exposure (NO$_{exp}$, in molecules cm$^{-3}$ s) divided by initial NO concentration, under various conditions.**"

R1.9) l. 275: Instead of "similar with those cases" it must read "similar to those cases".

Corrected as the Referee suggested.

R1.10) l. 394: "Despite its double bond, ethene reacts as slowly with NO3 as alkanes, likely due to lack of alkyl groups enriching electron density on the C=C bond, which slows NO3 addition." Why is this relevant here?

In that text we explained why ethene is different from other alkenes. Readers can thus get the message that NO$_3$ reacts rapidly with species with C=C bond, except ethene. It is not rigorous to draw the conclusion that alkenes react rapidly with NO$_3$ without mentioning this exception.

R1.11) l. 439-441: This sentence is confusing, the word "process" seems out of place here.

We rewrite the sentence as follows:

"**As a result of NO$_{3exp}$/OH$_{exp}$ ~100, only a minor portion of cresol could have undergone OH addition and then H-elimination again. This pathway leads to the formation of methyldihydroxybenzenes and other OH-oxidation products (Atkinson and Arey, 2003).**"

R1.12) l. 444-447: In this sentence, please briefly state again (maybe in parenthesis) which route is which in this example (H-abstraction vs. OH addition) to avoid confusion.

We have stated the pathways in parenthesis and the text in L444 now reads:

"**In summary, the model results suggest that there were two possible routes leading to nitroaromatic formation. However, one of them (recombination of OH-aromatic adducts with NO$_2$) is likely of little atmospheric relevance due to very high NO$_x$ needed, and the other (H-abstraction from cresol) occurs in the atmosphere but is not a major fate of aromatics (Calvert et al., 2002).**"

R1.13) l. 465: "… suppression can as high …" should read "… suppression can be as high… ".

Corrected as the Referee suggested.

R1.14) l. 477: "most hot stabilized period". Is there a word missing here?

The corrected sentence reads:

"**A dilution by a factor of 12, as actually used by Karjalainen et al. (2016), appears to be sufficient to bring most of the hot stabilized period under good conditions (Fig. S6).**"

R1.15) Fig. S1: please use consistent colors for chemical species.

Having double-checked, we think that colors for species in Fig. S1 are consistent. In all panels of Fig. S1, all the species concentrations (or concentration ratio) have one-to-one correspondence with line styles/colors.

**Anonymous Referee #2**

The current paper explores a chemical space extended to consider high NO concentrations within an OFR. Such a contribution, whilst of limited interest outside the immediate field, should be of considerable worth to users of such devices, particularly those looking to explore the emissions from high temperature combustion sources. However, to some degree, the paper is missing the same point that many previous theoretical characterisations of the devices also miss. The chemical space is just one element influencing the performance and atmospheric relevance of all PAM-type reactors (and the gas phase chemical space, just a subset of this). It is for this reason that I would hope that the current study is envisaged as one of a series of papers that will be extended to the dynamical, physical and condensed-phase chemical considerations. I will return to this below.

Having said this, within the stated scope, this paper carries a large amount of good new work that will make it worthy of publication in ACP. The chemical modelling appears appropriate with most of the necessary rate constants relatively well-constrained. This allows the characterisation of "good", "risky" and "bad" conditions under both 185 and 254 nm photolysis, though with the same caveats to the earlier work relating to uncertainties in the photolysis cross-sections and product yields of all possible VOCs (particularly when considering complex mixtures as in combustion emissions). In general, I am in agreement with the other referee that the gas phase chemical modelling alone warrants publication in ACP, but would invite the authors to address two main points to establish the validity of the approach and one point relating to the contextualisation of their study.

R2.1) Validity of the plug flow assumption: in section 3.1.3 it is stated that the uncertainties relating to kinetic parameters are relatively low compared to other factors including the plug flow assumption, referring to Peng et al., 2015. It would be useful for the authors to discuss whether the relative kinetic vs dynamic uncertainties under the high NOx conditions are of a comparable magnitude to those under low NOx conditions. There have been plenty of studies of axial and radial gradients in flow reactors, so some justification of the highly simplified modelling approach would be appropriate, given the biggest uncertainties are explicitly stated as being related to this assumption.

We have investigated the impacts of a residence time distribution (RTD) measured by Lambe et al. (2011). Under most conditions, the difference between $OH_{exp}$ from the plug-flow and RTD models is relatively small (within a factor of 3), while at high UV, $OHR_{ext}$, and $NO^{in}$, the difference can be larger. All main conclusions in this paper still hold after the discussions about the RTD impacts are included.

We have added Section 3.3 for discussion of RTD effects:

"**3.3 Effect of non-plug flow**

We performed model runs where the only change with respect to our box model introduced in Section 2.2 is that the plug-flow assumption is replaced by the residence time distribution (RTD) measured by Lambe et al. (2011) (also see Fig. S8 of Peng et al. (2015)). The chemistry of different air parcels with different residence times is simulated by our box model and outputs are averaged over the RTD. Lateral diffusion between different air parcels is neglected in these simulations.

$OH_{exp}$ calculated from the mode with RTD ($OH_{exp,RTD}$) is higher than that calculated from the plug-flow model ($OH_{exp,PF}$) in both OFR185-iNO and OFR254-iNO (Table 4 and Fig. S6). Under most explored conditions deviations are relatively small, which leads to an overall positive deviation of $OH_{exp,RTD}$ from $OH_{exp,PF}$ by ~x2 (within the uncertainties of the model and its application to real experimental systems). For OFR185-iNO, most conditions (~90%) in the explored space lead to <x3 differences between $OH_{exp,PF}$ and $OH_{exp,RTD}$, while for a small fraction of cases the differences can be larger (Fig. S6). The larger deviations are mainly present at high UV, $OHR_{ext}$, and $NO^{in}$, where conditions are generally "bad" and in which experiments are of little atmospheric relevance. Under these specific conditions, external OH reactants and $NO_y$ can be substantially destroyed for the air parcels with residence times longer than the average, while this is not the case for the average residence time. This feature was already described by Peng et al. (2015) (see Fig. S10 of that study). Although only non-$NO_y$ external OH reactants were considered in that study, the results are the same. In the present study, a higher upper limit of the explored $OHR_{ext}$ range (compared to Peng et al., 2015, due to trying to simulate extremely high $OHR_{ext}$ used in some recent literature studies) large amounts of $NO_y$ and cause somewhat larger deviations. In OFR254-iNO, OH is less suppressed at high $OHR_{ext}$ and $NO^{in}$ than in OFR185-iNO because of high $O_3$ (Peng et al., 2015), $OH_{exp,RTD}$ deviations from $OH_{exp,PF}$ are also smaller (Table 4).

Table 4. Statistics of the ratio between OH exposures calculated in the model with the Lambe et al. (2011) residence time distribution ($OH_{exp,RTD}$) and in the plug-flow model ($OH_{exp,PF}$). The geometric mean, uncertainty factor (geometric standard deviation), and percentage of outlier cases (>3 or <1/3) are shown for OFR185-iNO, OFR254-70-iNO, and OFR254-7-iNO.

|  | Geometric mean | Uncertainty factor | Outlier cases (%) |
| --- | --- | --- | --- |
| OFR185-iNO | 1.91 | 1.64 | 11 |
| OFR254-7-iNO | 1.59 | 1.51 | 7 |
| OFR254-70-iNO | 1.48 | 1.29 | 3 |

[Figure]

**Figure S6. Scatter plot of OH exposure calculated in the model with the Lambe et al. (2011) residence time distribution ($OH_{exp,RTD}$) vs. that calculated in the plug-flow model ($OH_{exp,PF}$) for OFR185-iNO, OFR254-7-iNO, and OFR254-70-iNO. 1:1, 1:3, and 3:1 lines are also shown for comparison.**

**Based on the outputs of the model with RTD, similar mapping of the physical input space as Figs. 4 and 5 can be done (Figs. S7 and S8). Overall, the mapping of the RTD model results is very similar with that of the plug-flow model. The conditions appear to be only slightly better in a few places of the explored space than those from the plug-flow model, which can be easily explained by the discussions above. Besides, the mapping in Figs. S7 and S8 also appear to be slightly more low-NO, for the same reasons discussed above. After NO is destroyed at long residence times, $HO_2$, suppressed by NO, also recovers as OH. $r(RO_2+NO)/r(RO_2+HO_2)$ is obviously expected to be smaller than in the plug-flow model in general.**

[Figure]

Increasing UV

Increasing H₂O

**Case LL**      **Case LM**      **Case LH**

**Case ML**      **Case MM**      **Case MH**

**Case HL**      **Case HM**      **Case HH**

**Figure S7. Same format as Fig. 4, but for the OFR185-iNO results obtained by the model with the Lambe et al. (2011) residence time distribution.**

[Figure]

**Figure S8.** Same format as Fig. 5, but for the OFR254-22-iNO results obtained by the model with the Lambe et al. (2011) residence time distribution.

**Note that most conditions that appear to be better in the RTD model results are already identified as bad by the plug-flow model. Those conditions look slightly better only because of their better *RTD-averaged* F185$_{exp}$/OH$_{exp}$ and F254$_{exp}$/OH$_{exp}$. However, each of those cases is actually composed of both a better part at longer residence times and also a worse part at shorter residence times. Under those conditions, the reactor simultaneously works in two distinct regimes, one of which is bad due to heavy OH suppression. Such conditions are obviously not desirable for OFR operation.**"

R2.2) Validity of separating the numerical treatment of gas phase and particle phase processes: there is no statement of any of the uncertainty in gas phase chemistry being attributable to multiphase processes. I find this rather curious, since the primary focus of most PAM chamber studies relates is particulate mass. Both radical and closed shell species may interact substantially with the particle phase. All the particulate material in SOA particles is, by definition, formed from the vapour phase. If the flow regime is anything near plug flow, then the particle number, condensation sink, mass and composition of the particulate will evolve with the gas phase species and hence mass transfer (in both directions, where there is oxidative fragmentation and functionalisation) will be changing temporally and spatially inside the reactor. There really should be some discussion of the potential impacts of these processes in the paper.

We believe that separation of gas-phase and particle phase processes can only have minor impacts on both gas-phase and particle-phase chemistries in OFR and is thus a valid approximation.

We have modified the text to L144 to provide some discussion of this issue:

**"As in Peng et al. (2015, 2016), SO$_2$ is used as a surrogate of external OH reactants (e.g., VOCs). NO$_y$ species, although also external OH reactants, are explicitly treated in the model and *not* counted in OHR$_{ext}$ in this work. Therefore, OHR$_{ext}$ stands for *non*-NO$_y$ OHR$_{ext}$ only hereinafter, unless otherwise stated. Also, particle-phase processes and interactions of gas-phase species with particles are not considered in this study. We have made this assumption because:**

**i)      The presence of aerosols has typically negligible impacts on the gas-phase chemistry. Condensational sink (CS) of ambient aerosols can rarely exceed 1 s$^{-1}$ even in polluted areas and is usually 1-3 orders of magnitude lower (Donahue et al., 2016; Palm et al., 2016). Thus, even under the assumption of unity uptake coefficient, CS cannot compete with OHR$_{ext}$ (usually on the order of 10 s$^{-1}$) in OH loss. Uptake of NO onto aerosols only occurs through the reaction with RO$_2$ on particle surface (Richards-Henderson et al., 2015), which is formed very slowly (see below) compared to gas-phase HO$_x$ and NO$_x$ chemistry. Uptake of HO$_2$, O$_3$, NO$_3$ etc. is even more unlikely to be efficient (Moise and Rudich, 2002; Moise et al., 2002; Hearn and Smith, 2004; Jathar et al., 2016).**

**ii)** **On the other hand, gas-phase species have only limited impacts on OA. Heterogeneous oxidation of OA by OH is generally slow. Significant OA loss due to heterogeneous oxidation can only be seen at photochemical ages as high as weeks (Hu et al., 2016). The enhancement of heterogeneous oxidation due to NO is remarkable only at OH concentration close to the ambient values but not at typical values in OFR (Richards-Henderson et al., 2015).**

**It is an important approximation that the *real* $OHR_{ext}$ decay (due to not only primary VOC oxidation and subsequent oxidation, but also wall loss, partitioning to the particle phase, reactive uptake etc.) is surrogated by that of $SO_2$. Gas-phase measurements in literature laboratory studies revealed that there is a large variability of total $OHR_{ext}$ during (subsequent) oxidation of VOCs, depending on the type of precursors (Nehr et al., 2014; Schwantes et al., 2017). This variability is obviously mainly due to the evolution of different types of oxidation intermediates/products contributing to $OHR_{ext}$, but not due to changes in CS, wall conditions etc. Also this variability is difficult to accurately capture even if modeling with a mechanism as explicit as MCM is performed (Schwantes et al., 2017). It is thus justified to use a lumped surrogate to model the $OHR_{ext}$ decay for simplicity and efficiency. The uncertainties introduced by this approximation include those due to both the types of oxidation intermediates/products and all interactions of VOCs with aerosols, walls etc. And the uncertainties due to the former dominate over those due to the latter."**

R2.3) My final point relates to the context of the study. If it is not envisaged that this second paper on the chemical characterisation of OFRs is to eventually be accompanied by a numerical study of the multiphase processes, then I think the paper requires quite a bit more contextualisation. The root of the missing material relates to the competition between processes (nucleation, condensation, evaporation, coagulation, condensed phase reaction) alluded to in point ii) above and relating to aerosol dynamical evolution that are highly dependent on the magnitudes of different moments of the aerosol distribution. Extrapolation to concentration regimes other than the dilutions under the operating conditions of the OFR is simply not possible without the adoption of substantial questionable assumptions or use of a highly complex model which has yet to be described. The current paper implicitly aims to limit its scope to gas phase oxidation of VOCs in the OFR, but this is seldom the purpose to which they are put. Indeed, the limited context for OFR studies explicitly points to their use for "...secondary organic aerosol (SOA) formation and aging [studies], in both the laboratory and the field", because of the perceived advantage of elevated oxidant levels. None of the disadvantages that are related directly to the inappropriate exptrapolation of all the multiphase processes of relevance to SOA formation and transformation are mentioned. This requires significant rebalancing, ideally quantitatively in a further detailed publication but at least qualitatively in the introduction of the current paper.

First of all, a reactor such as an OFR is complex and can involve gas, heterogeneous, particle-phase chemistry, gas-particle partitioning thermodynamics and kinetics, size distribution dynamics, three-dimensional flow fields and UV light distributions, different wall materials, and small temperature non-uniformities in some cases. In addition, an OFR can be used in a multitude

of configurations and input conditions. It is impossible to investigate all the processes in a single paper, especially when some of the processes (e.g. the impact of high initial NO in the gas-phase chemistry in OFRs) had never been investigated before. Our approach has been to tackle important parts of the overall phase space in individual papers. In particular we are focusing on the gas-phase chemistry in several of our papers because (1) there seems to be limited understanding of it in the OFR community, (2) at least some literature studies may have been conducted under conditions far from atmospheric relevance; and (3) once this chemistry is understood, there are relatively easy and practical ways to plan experiments to avoid major problems, and to quantify the relative effects of different processes. We are working on additional manuscripts and we hope to continue to be active in this area, but overall OFR modeling is a subfield in itself, and our group cannot be expected to address every single possible topic. Even for environmental chambers, which have been around for over 6 decades, very few modeling publications consider the gas and particle chemistry and size distribution dynamics simultaneously.

Importantly, we would like to let the Referee know that we are currently collaborating with the group of Jeffrey Pierce at Colorado State University on detailed aerosol dynamics modeling in OFR, including nucleation, condensation, and coagulation, as well as heterogeneous chemistry, and our collaborators have already presented some preliminary results (Hodshire et al., 2017).

As stated in Hu et al. (2016), "the OFR does not accelerate processes such as aerosol uptake and reactions that do not scale with OH". This feature of OFR is rather straightforward. None of aerosol dynamical processes except the uptake of species with elevated concentrations (OH, $HO_2$ etc.) relative to those in the atmosphere are enhanced in OFR. The short residence times and high LVOC production rates may prevent SOA growth from reaching equilibrium (Palm et al., 2016; Ahlberg et al., 2017). Also, common particle-phase chemical reactions (e.g., carbonyl-amine browning (Haan et al., 2009) and cyclic hemiacetal formation and dehydration (Strollo and Ziemann, 2013)) do not involve OH and are not accelerated in OFR. Heterogeneous OA oxidation by OH is accelerated but its main pathways are identical to those in the gas phase (Houle et al., 2015; Richards-Henderson et al., 2015) and is not as important as the gas-phase radical chemistry in terms of species production and consumption amounts (see response to R2.2). Therefore, we had not intended to limit the scope of this paper within the gas phase. As the title of this paper reads, we focus on OFR chemistry with NO, but for the reasons above, a gas-phase model is sufficient to investigate the main features of this chemistry.

Most importantly, "atmospheric relevance" in this paper does not refer to a perfect reproduction of all processes of interest in the atmosphere, as none of the reactors used for atmospheric chemistry and aerosol research can achieve this. We aim to understand the chemistry in the reactor to enable us and others to avoid the processes that do not occur in the atmosphere, and to understand the deviations in the relative importance of the processes that do occur. In OFR, aerosol dynamics may be relatively slower, compared to accelerated reactions with OH, even though both occur in both OFR and the atmosphere. Specific input conditions and/or measures of intervention may be adopted to modify and/or investigate such issues. For instance, pure sulfuric acid particles may be used to enhance the reactive uptake of IEPOX (Hu et al., 2016); or

seed particles may be injected to avoid over-oxidation of LVOCs in the gas phase before their condensation onto particles (Palm et al., 2016).

In summary, we believe that the scope of this paper and the use of a gas-phase model in this paper are appropriate, and a detailed investigation of particle-related processes, which is on-going, will result in a future paper.

R2.4) Related to the above, the previous findings of nitrogen being incorporated into SOA are very tricky to interpret. The recommendations for operation are made from the perspective of gas phase oxidation to ensure that the gas phase product distribution is not anomalous. Incorporation of the nitrogenous species into particles will be subject to multiphase processes leading to net mass transfer between the phases. The mass transfer rate will be proportional to not only the difference between the gaseous concentration and the equilibrium concentration above the particle, but also to the condensation sink provided by the particles. Extrapolation to the amount of a component or class of components in the SOA (e.g. nitrogen-containing ones) to ambient conditions should not only consider the equivalent oxidant dose and gas phase chemical regime, but also aim to establish some equivalence in terms of the mass transfer between phases.

To our knowledge, Liu et al. (2015) is the only published OFR study reporting the incorporation of nitrogen into SOA. Their interpretation of this observation did *not* involve multiphase *chemical* processes. They interpreted their nitrogen-containing compounds observed in SOA as organic nitrates formed by $RO_2$+NO and nitroaromatics formed by reactions of phenoxy with $NO_2$. Both pathways have been extensively discussed in our paper. In addition, we have found by modeling that under similar conditions with theirs, recombination of OH-aromatic adducts with $NO_2$ can be faster than that with $O_2$. Since OH-aromatic adducts can be the products of the very first step of aromatic (SOA precursors in that study) oxidation, nitroaromatic formation via this pathway may be substantial (see Section 3.3.3 in the ACPD paper). All abovementioned pathways are gas-phase reactions. The products may undergo further oxidation till their volatilities are sufficiently low to condense onto aerosols.

Although it cannot be ruled out, nitrogen incorporation due to reactive uptake of NO leading to organic nitrates formation in the particle phase was found to be negligible (Richards-Henderson et al., 2015). Therefore we do not agree with the Referee that complex multiphase mass transfer considerations are necessary to interpret nitrogen incorporation into OA, at least from current experimental reports.

**References (for responses to both reviewers)**

Ahlberg, E., Falk, J., Eriksson, A., Holst, T., Brune, W. H., Kristensson, A., Roldin, P. and Svenningsson, B.: Secondary organic aerosol from VOC mixtures in an oxidation flow reactor, Atmos. Environ., 161, 210–220, doi:10.1016/j.atmosenv.2017.05.005, 2017.

Atkinson, R. and Arey, J.: Atmospheric degradation of volatile organic compounds., Chem. Rev., 103(12), 4605–38, doi:10.1021/cr0206420, 2003.

Calvert, J. G., Atkinson, R., Becker, K. H., Kamens, R. M., Seinfeld, J. H., Wallington, T. H. and Yarwood, G.: The Mechanisms of Atmospheric Oxidation of the Aromatic Hydrocarbons, Oxford University Press, USA. [online] Available from: https://books.google.com/books?id=P0basaLrxDMC, 2002.

Donahue, N. M., Posner, L. N., Westervelt, D. M., Li, Z., Shrivastava, M., Presto, A. A., Sullivan, R. C., Adams, P. J., Pandis, S. N. and Robinson, A. L.: Where Did This Particle Come From? Sources of Particle Number and Mass for Human Exposure Estimates, in Airborne Particulate Matter: Sources, Atmospheric Processes and Health, edited by R. M. Harrison, R. E. Hester, and X. Querol, pp. 35–71, Royal Society of Chemistry., 2016.

Haan, D. O. De, Corrigan, A. L., Smith, K. W., Stroik, D. R., Turley, J. J., Lee, F. E., Tolbert, M. A., Jimenez, J. L., Cordova, K. E. and Ferrell, G. R.: Secondary Organic Aerosol-Forming Reactions of Glyoxal with Amino Acids, Environ. Sci. Technol., 43(8), 2818–2824, doi:10.1021/es803534f, 2009.

Hearn, J. D. and Smith, G. D.: Kinetics and Product Studies for Ozonolysis Reactions of Organic Particles Using Aerosol CIMS †, J. Phys. Chem. A, 108(45), 10019–10029, doi:10.1021/jp0404145, 2004.

Hodshire, A., Palm, B., Jimenez, J.-L., Bian, Q., Pierce, J. R., Campuzano-Jost, P., Day, D., Peng, Z., Ortega, A., Hunter, J., Cross, E., Kroll, J., Kaser, L., Jud, W., Karl, T. and Hansel, A.: Nucleation and Growth Under High Oh Conditions: Using an Oxidation Flow Reactor and the Tomas Box Model to Learn About Chemistry, Nucleation, and Growth Potential of Ambient Pine-Forest Air, in American Association for Aerosol Research 36th Annual Conference, p. Abstract Number: 59. [online] Available from: http://aaarabstracts.com/2017/viewabstract.php?pid=59, 2017.

Houle, F. A., Hinsberg, W. D. and Wilson, K. R.: Oxidation of a model alkane aerosol by OH radical: the emergent nature of reactive uptake, Phys. Chem. Chem. Phys., 17(6), 4412–4423, doi:10.1039/C4CP05093B, 2015.

Hu, W., Palm, B. B., Day, D. A., Campuzano-Jost, P., Krechmer, J. E., Peng, Z., de Sá, S. S., Martin, S. T., Alexander, M. L., Baumann, K., Hacker, L., Kiendler-Scharr, A., Koss, A. R., de Gouw, J. A., Goldstein, A. H., Seco, R., Sjostedt, S. J., Park, J.-H., Guenther, A. B., Kim, S., Canonaco, F., Prévôt, A. S. H., Brune, W. H. and Jimenez, J. L.: Volatility and lifetime against OH heterogeneous reaction of ambient isoprene-epoxydiols-derived secondary organic aerosol (IEPOX-SOA), Atmos. Chem. Phys., 16(18), 11563–11580, doi:10.5194/acp-16-11563-2016, 2016.

Jathar, S. H., Cappa, C. D., Wexler, A. S., Seinfeld, J. H. and Kleeman, M. J.: Simulating secondary organic aerosol in a regional air quality model using the statistical oxidation model –

Part 1: Assessing the influence of constrained multi-generational ageing, Atmos. Chem. Phys., 16(4), 2309–2322, doi:10.5194/acp-16-2309-2016, 2016.

[revised manuscript text omitted]

---

## Author Response (AR2)

**Second round review**

We thank the Referee #2 again for his review. To facilitate the review process we have copied the reviewer comments in black text. Our responses are in regular blue font. We have responded to all the referee comments and made alterations to our paper (**in bold text**). Page and line numbers refer to the *first revised* manuscript.

**Referee #2**

The authors have largely addressed my concerns in their rebuttal. I am happy that the paper can form the basis of ongoing discussions in the field of OFR deployment and interpretation. I am particularly satisfied with the RTD analysis and its explanation of the deviation from plug flow. I am still slightly concerned by the continued separation of the gaseous and condensed phase processes. I am in full agreement with the authors that "OFR modeling is a subfield in itself, and our group cannot be expected to address every single possible topic". However, where a process can have a substantial influence on the processes that are the subject of a manuscript, then this possibility should be acknowledged.

It can be argued that the two statements in point 2.2 of the authors response:

i) "The presence of aerosols has typically negligible impacts on the gas-phase chemistry" and ii) ..."gas-phase species have only limited impacts on OA"

are not demonstrably correct for all conditions in OFRs.

R2.5) To rebut i), consider the typical concentrations in diesel emissions. Concentrations of NOx in raw diesel exhaust are typically between 50 and 1000 ppm depending on running conditions and technology (and can be very much higher during transients and below 17 degrees C when EGR is not mandated). Clearly this is the sort of NOx target regime of the current manuscript. Whilst PM emissions do not respond in the same way as NOx to engine technologies (e.g. EGR generally increases PM whilst decreasing NOx, and only DPF fitted vehicles have significantly reduced PM) or load-speed conditions, typical concentrations from a modern light-duty (EURO5) diesel generally range from between 1 and 30 mg/m^3 in raw exhaust. Assuming 80 nm modal diameter, 1 mg/m^3 will provide a mass transfer rate ("condensation sink") of about 4 s^-1 (using an uptake coefficient of unity); so a lifetime of 0.25 s for such a condensing gas (and 30 mg/m^3 would give a lifetime of less than 0.01 s). A lower uptake coefficient would obviously lead to a longer lifetime (e.g. 1 s for 0.01 at 30 mg/m^3).

Lines 88 to 92 explicitly include OFR conditions where there is a substantial likelihood of such high primary PM mass (an urban tunnel, "where NOx was high enough to be a major OH reactant"... and ... emissions of vehicles, biomass burning, and other combustion sources, "where NO can often be hundreds of ppm"). Looking at the Karjalainen et al., 2016 case presented in Figures 7 and 8, the authors are carrying out calculations under raw, 12 x and 100 x dilution conditions for gasoline engine emissions. Figure 7 in Karjalainen reported average primary PM values of 0.45 mg/m^3 for parts of the test cycle (assumed raw), rising to more than 10 mg/m^3 including the SOA from a gasoline engine. Similarly, the Link et al., 2016 study of diesel emissions at 45 - 110 dilution employed no primary particle removal technology to emissions from a turbocharged, intercooled, heavy-duty, off-road diesel engine likely to emit massively more than the light-duty levels stated above (in excess of 100 mg/m^3 is readily possible in raw exhaust from such engines). In both these cases the mass transfer of potentially condensing closed shell and radical species to PM could clearly provide very significant sinks of gaseous components that should be considered in a model of OFRs.

We realize that the Referee is not understanding what we intended to communicate with our statements. When we stated that the "gas-phase chemistry" was not significantly perturbed by the presence of particles, we were referring to the gas-phase chemistry that we are modeling, i.e., the radical and $NO_x$-$NO_y$ chemistry, as well as the consumption of VOCs and other OH reactants. We did not intend to include the physical partitioning of semivolatile and low volatility species, which clearly is an area of strong interaction of the gas and particle phases (but for the most part does not involve chemistry). We will clarify the language, as described below, to remove this potential source of confusion.

We thank the Referee for providing detailed examples for high condensational sink in combustion exhausts. Nevertheless, even if raw exhausts are injected into the reactor, they still cannot have significant impacts on the major gas-phase oxidants, since VOCs and $NO_x$ in raw exhausts, which are also proportionally higher, still dominate total oxidant sink. Dilution of raw exhausts simultaneously lowers condensational sinks and gas-phase reactive oxidant sinks, with their relative importance remaining the same.

We have added a few sentences to clarify this point at the end of the paragraph i) in L151. The modified paragraph now reads:

i) **"The presence of aerosols has typically negligible impacts on the gas-phase chemistry of radicals, $NO_x$/$NO_y$, and OH reactants studied here. Condensational sink (CS) of ambient aerosols can rarely exceed 1 s$^{-1}$ even in polluted areas and is usually 1-3 orders of magnitude lower (Donahue et al., 2016; Palm et al., 2016). Thus, even under the assumption of unity uptake coefficient, CS cannot compete with OHR$_{ext}$ (usually on the order of 10 s$^{-1}$ or higher) in OH loss. Uptake of NO onto aerosols only occurs through the reaction with RO$_2$ on particle surface (Richards-Henderson et al., 2015), which is formed very slowly (see below) compared to gas-phase HO$_x$ and $NO_x$ chemistry. Uptake of HO$_2$, O$_3$, NO$_3$ etc. is even more unlikely to be of importance due to lower uptake coefficients (Moise and Rudich, 2002; Moise et al., 2002; Hearn and Smith, 2004; Lakey et al., 2015). Combustion exhausts can have high aerosol loadings with condensational sinks on the order of 10$^2$–10$^3$ s$^{-1}$ (Matti Maricq, 2007). Even if these exhausts are directly injected into the reactor without any pre-treatment, uptake onto the particles still cannot play a major role in the fate of gas-phase radical and $NO_x$ species, since VOCs and $NO_x$ in raw exhausts, which are**

**proportionally orders-of-magnitude higher, still dominate the fate of oxidants. Dilution of combustion emissions simultaneously lowers condensational sinks and the sinks of oxidants due to chemical reactions, with their relative importance remaining the same as in undiluted emissions."**

We acknowledge that a strong dilution leads to a lower PM loading than in raw exhausts, which slows down the uptake. But this "lower" PM loading may still be much higher than typical ambient values after dilution. So it may not necessarily be "low PM loading" compared to ambient conditions, as suggested by the Referee below (R2.8).

We have added some text to L566 to acknowledge the PM loading change by dilution:

"**Note that a strong dilution lowers aerosol mass loading in vehicle emissions. As a result, condensation of gases onto particles is slower than in raw exhausts. However, condensational sinks after dilution may still be significantly higher than typical ambient values (Matti Maricq, 2007; Donahue et al., 2016).**"

R2.6) To address author response 2.2 ii), clearly gas-phase species have a strong impact on OA, being 100% responsible for all SOA. Mass transfer of semi-volatile and low volatility gas-phase species (in the case of exhaust experiments, almost completely due to condensation on existing primary PM) has a determinant effect on PM mass. Gas phase oxidants may have a limited impact on OA chemistry, but gas phase species have a profound effect on OA. Given the paper title relates to modelling the chemistry in OFRs (not modelling the oxidants), it is not solely transfer of radical species between phases that is of concern.

As discussed in response to R2.5, we agree with this and did not intend to say otherwise. However this mostly concerns physics ("mass transfer" in the Referee's words) and not chemistry. We have modified that paragraph to clarify the fact that physical uptake of semivolatile and low-volatility gas-phase species have a strong impact on PM mass:

ii)      "**Gas-phase radical and $NO_x/NO_y$ species only has limited impacts on OA *chemistry* in this study. Heterogeneous oxidation of OA by OH is generally slow. Significant OA loss due to heterogeneous oxidation can only be seen at equivalent photochemical ages as high as weeks (Hu et al., 2016). The enhancement of heterogeneous oxidation due to NO is remarkable only at OH concentration close to the ambient values but not at typical values in OFR (Richards-Henderson et al., 2015).**

**It is well known that the aerosol concentration can have a major impact on the physical uptake of semivolatile and low-volatility gas-phase species. However this process is not explicitly modeled in this study.**"

Also, we have modified text to L148 for more clarity:

"**Also, particle-phase chemistry and physical and chemical interactions of gas-phase species with particles are not considered in this study.**"

R2.7) In the context of the above discussion, I do not understand the final paragraph of the authors suggested added text in point 2.2. I think this requires further explanation before inclusion in the paper.

The core idea of that paragraph is that the details of VOC product evolution have a strong influence on the rate of OH loss, but this is highly complex and not well captured even by models as explicit as MCM. We surrogated VOC evolution by $SO_2$ for simplicity and efficiency and this introduces much more uncertainty on (the temporal variation of) $OHR_{ext}$ due to VOC (and hence the radical chemistry) than mass transfer to the particle phase does. Thus, there is no strong need for explicitly including gas-particle mass transfer in the present modeling work.

We have modified that paragraph (L166) below for more clarity:

"**As $OHR_{ext}$ plays a major and even dominant role in OH loss, it is an important approximation that the *real* $OHR_{ext}$ decay (due to not only primary VOC oxidation and subsequent oxidation of higher generation products, but also wall loss, partitioning to the particle phase, reactive uptake etc.) is surrogated by that of $SO_2$ (see Fig. S2 of Peng et al. 2015). Gas-phase measurements in literature laboratory studies revealed that there is a large variability of the evolution of total $OHR_{ext}$ during oxidation of primary VOCs and subsequent oxidation of their intermediate products, depending on the type of precursors (Nehr et al., 2014; Schwantes et al., 2017). This variability is obviously mainly due to the formation of different types and amounts of oxidation intermediates/products contributing to $OHR_{ext}$. This variation is highly complex due to the large number of possible oxidation intermediates and the limited knowledge of detailed higher-generation mechanisms, and thus is difficult to accurately capture even if modeling with a mechanism as explicit as Master Chemical Mechanism is performed (Schwantes et al., 2017). Therefore, it is justified to use a lumped surrogate to model the $OHR_{ext}$ decay for simplicity and efficiency. This approximation is a major contributor to uncertainty of our model. The uncertainties due to both the types of oxidation intermediates/products.**"

R2.8) Whilst I do not expect the paper to explicitly address coupling of the gaseous and particulate processes, I would expect the current manuscript to at least acknowledge the interaction between the gas phase chemistry and gaseous losses associated with condensation and the resultant increase in PM mass. The authors should state that their study is completely relevant for low PM loadings in OFRs, but care should be taken when applying it to high ambient PM concentration or direct emission studies (both raw and diluted). Clearly the authors are aware of the necessity to include coupled multiphase processes and should be commended in their work with Jeff Pierce's group on this.

We have acknowledged (see the response to R2.5) that dilution changes PM loadings and hence mass transfer rates, and, when diluted, sources have lower PM loadings in OFRs than in raw emissions. However, we believe that the statement that our study is only valid to low PM loadings is incorrect, as discussed in detail in our response to R2.5. Thus no changes have been made in response to this point.

[revised manuscript text omitted]